# The atlas of unburnable oil for supply-side climate policies

Lorenzo Pellegrini [1,5], Murat Arsel [1], Gorka Muñoa [2,3], Guillem Rius-Taberner [2,3], Carlos Mena[4] & Martí Orta-Martínez [2,3,5] ✉

To limit the increase in global mean temperature to 1.5 °C, $CO_2$ emissions must be drastically reduced. Accordingly, approximately 97%, 81%, and 71% of existing coal and conventional gas and oil resources, respectively, need to remain unburned. This article develops an integrated spatial assessment model based on estimates and locations of conventional oil resources and socio-environmental criteria to construct a global atlas of unburnable oil. The results show that biodiversity hotspots, richness centres of endemic species, natural protected areas, urban areas, and the territories of Indigenous Peoples in voluntary isolation coincide with 609 gigabarrels (Gbbl) of conventional oil resources. Since 1524 Gbbl of conventional oil resources are required to be left untapped in order to keep global warming under 1.5 °C, all of the above-mentioned socio-environmentally sensitive areas can be kept entirely off-limits to oil extraction. The model provides spatial guidelines to select unburnable fossil fuels resources while enhancing collateral socio-environmental benefits.

The Paris Agreement aims to limit global warming to well below 2 °C, pursuing efforts to limit it to 1.5 °C, compared to pre-industrial levels. Since global temperature rise is closely related to cumulative anthropogenic $CO_2$ emissions, the remaining carbon budget is the total amount of $CO_2$ emissions that should not be exceeded to limit average global warming to the desired temperature[1–4]. For a 50% probability of limiting warming to 1.5 °C, the remaining carbon budget from 2020 onwards has been estimated at 500 gigatonnes (Gt) of $CO_2$ (400 $GtCO_2$ for a 67% probability)[5]. For a 2 °C limit, the remaining carbon budget amounts to 1350 $GtCO_2$ for a 50% probability (1150 $GtCO_2$ for a 67% probability)[5]. These budgets are continuously depleted at current rates of emissions of about 42 $GtCO_2$ per year[6] and the one concerning 1.5 °C warming could be completely exhausted by 2030[7]. Moreover, the unabated exploitation of all the world's existing fossil fuel resources is incompatible with achieving the Paris Agreement goals since the $CO_2$ combustion emissions of global fossil fuel resources (nearly 11,000 $GtCO_2$) are 22 times higher than the 1.5 °C carbon budget[1,8].

The discrepancy between the carbon budget and the $CO_2$ emissions embedded in global fossil fuel resources has profound implications for the future of global energy consumption[9] and has prompted a discussion on 'unburnable fuels' (or 'unextractable') and the associated risk of stranded assets[10,11]. There is currently a surge in interest by academics and policy-makers on supply-side climate policies to limit fossil fuel production (e.g., extraction taxes, fossil fuel subsidy removal, moratoria or quotas on extraction, tradable production allowances, and restrictions on the access to credit)[12–14]. Several countries have unilaterally started various types of (partial) moratoria[12,15–17]. Social movements have been targeting specific fossil fuel extraction or infrastructural projects intending to eliminate them altogether[18,19]. Proposals have been made for a non-proliferation fossil fuel treaty[20] and the Beyond Oil and Gas Alliance, the first global coalition of governments committing to the managed phase-out of oil and gas production[21], has been established. Initiatives and campaigns at the intersection between academia and activism, such as Leave It in the

[1]International Institute of Social Studies (ISS), Erasmus University Rotterdam, The Hague, the Netherlands. [2]Departament de Biologia Evolutiva, Ecologia i Ciències Ambientals, Facultat de Biologia, Universitat de Barcelona, Barcelona, Catalonia, Spain. [3]Institute de Recerca de la Biodiversitat (IRBio), Universitat de Barcelona, Barcelona, Catalonia, Spain. [4]Institute of Geography, Universidad San Francisco de Quito, Quito, Ecuador. [5]These authors contributed equally: Lorenzo Pellegrini, Martí Orta-Martínez. ✉e-mail: Marti.Orta@ub.edu

Ground (LINGO) and Oil Change International, have promoted further awareness of supply side climate policies. Finally, the International Energy Agency has acknowledged that a rapid decline of investment in fossil fuel extraction is needed and that several existing projects will have to be retired before they reach the end of their 'technical lifetime'[22].

The selection of the resources that need to stay under the ground vis-à-vis those that can be extracted is a crucial step on the way to imagining and constructing an effective international system to leave a share of existing global fossil fuel resources unextracted[8,15,20,23,24]. The allocation of the remaining fossil fuels that can be extracted is a morally and politically contentious issue[25] that is entwined with the issue of compensation[26]. In fact, (partial) compensation could be considered a condition for the political feasibility of any multilateral international agreement to keep fossil fuels unextracted[27] and the pathway to an agreement could include intermediary steps to strengthen international norms against fossil fuels[28] and club arrangements to pave the way to the agreement[29]. When it comes to the principles that can inform the distribution of fossil fuels that cannot be extracted, the global North (developed countries) has historically contributed the most to accumulated greenhouse gas emissions generating an ecological debt[30] and fossil fuel rents could contribute towards the global South's (developing countries) right to development[25]. Previous studies suggest, based on ethical considerations (philosophical principles of justice and fairness), that developed countries are those who should leave their fossil fuels underground, whereas studies based on economic efficiency suggest that those resources whose extraction generates lower rents (because of relatively higher extraction, transportation and transformation costs, or lower economic value of the resources) should be left untapped[31–33]. In this paper, we take a different approach and use the socio-environmental impacts of fossil fuel extraction in different locations to suggest alternative distributions of unburnable fossil fuels.

The only attempts to provide a global spatial distribution of unburnable fossil fuel resources have been at the continental scale based solely on the production costs of the different fossil fuel resources[8,23]. Welsby et al.[8] found that 71% and 81% of the conventional oil and gas resources, respectively, should remain unextracted by 2050[34]. 99% of unconventional oil resources, 93% of unconventional gas resources and 97% of coal should remain unburned by 2050 in order to limit global warming at the lowest overall cost[8]. According to Welsby et al.[8], estimates of unextractable conventional oil and gas resources by 2100 are reduced to 54% and 76%, respectively[8]. It is important to note that continued use of fossil fuels after 2050 depends also on Carbon Dioxide Removal (CDR). The research on criteria for the allocation of unburnable carbon has also focused on equity approaches but has not produced spatially explicit allocations of unburnable fossil fuels[31–33,35,36].

The dearth of specific proposals on the geospatial distribution of unburnable fossil fuels at the global level stands in contrast with local initiatives to leave untapped (specific) fossil fuel resources that overlap highly biodiverse areas or coincide with outstanding socio-environmental characteristics (e.g., the Yasuní-ITT proposal from the Ecuadorian Amazon, the Costa Rican moratorium on oil drilling and similar bans on fossil fuel projects in Alaska, Belize, and Mexico)[12,15,37]. Because of the local socio-environmental impacts of extraction, constraining fossil fuel supply can generate additional sustainability benefits[38–40]. Fossil fuel extraction has profound and enduring negative impacts on biodiversity[38,41,42] and potentially severe adverse effects on health and human rights[43–46]. Oil and gas extraction, in particular, can result in widespread terrestrial, marine and air pollution from spills[47], from gas flaring[48] and from the disposal of produced water[49], the main waste product of the oil and gas extraction industry. Produced water, found in the same formations with oil and gas, is brought to the surface during operations and can contain several toxic compounds, such as polycyclic aromatic hydrocarbons (PAHs), heavy metals and naturally occurring radioactive material[50]. The intended or accidental discharge of oil and produced waters and gas flaring have adverse public health effects[45,46,51]. In terms of direct social impacts, fossil fuel extraction has been found to increase the incidence of numerous social ills[17,52–55], and has also led to cases of contact with Indigenous Peoples living in voluntary isolation, with ensuing outbreaks of new diseases and high mortality rates[56,57]. The evidence regarding the impacts of fossil fuel extraction provides additional rationale for their conservation and offer the opportunity to choose specific fossil fuel resources to be kept untapped in order to maximize the collateral benefits of climate policies.

Here, we propose a methodology to identify and prioritize unburnable fossil fuel resources, apply it to the case of conventional oil and produce an atlas of unburnable conventional oil according to environmental and social criteria. The same methodology could be used to prepare atlases for unburnable coal and natural gas. We focus on conventional oil since almost all unconventional oil resources (1518 Gbbl) should remain unburned because extraction costs are much higher, tend to be less energy efficient (with lower Energy Return on Investment /EROI) and have more environmental impacts[8,58].

## Results
### The distribution of unburnable oil resources
To prioritize unburnable resources, we first assess their global distribution. Oil is usually categorized as 'conventional' or 'unconventional'. We follow Welsby et al.[8] that defines oil with density lower than water (often standardized as '10° API') as conventional (i.e., oil, light tight oil (LTO), and natural gas liquids (NGL)) and the remaining oil resources as unconventional (i.e., natural bitumen, extra-heavy oil, and kerogen oil). However, it is important to note that world energy institutions (i.e., United States Geological Survey (USGS), Society of Petroleum Engineers (SPE), U.S. Energy Information Administration (EIA) and International Energy Agency (IEA)) define LTO as unconventional—since LTO does not flow without stimulation and requires specialized extraction technology (i.e., hydraulic fracturing)[59–62].

Our georeferenced estimates of conventional oil resources are presented in Fig. 1. They show that the global spatial distribution of conventional oil resources is uneven and, out of the total of 2276 Gbbl, the largest amounts are in the sedimentary basins of Middle East (648 Gbbl or 28%; mainly in the Mesopotamian Foredeep Basin, 267 Gbbl, Zagros Fold Belt, 117 Gbbl, Greater Ghawar Uplift, 96 Gbbl and Rub Al Khali Basin, 101 Gbbl), the United States (402 Gbbl or 18%; mainly in the Permian Basin, 181 Gbbl, Gulf Coast Basins, 64 Gbbl, Northern Alaska, 36 Gbbl and, Appalachian Basin, 34 Gbbl), Russia and former Soviet states (343 Gbbl, or 15%; mainly in the West Siberia Basin, 189 Gbbl), Gulf of Guinea (West Central Coast, 61 Gbbl, and Niger Delta, 43 Gbbl), off-shore Brazil (Santos basin, 66 Gbbl, Campos Basin, 24 Gbbl), North Africa (Sirte Basin, 46 Gbbl, Trias/Ghadames Basin, 24 Gbbl) and the North Sea (33 Gbbl). However, oil is present in all continents and some of the most biodiverse regions have substantial portions of their area coinciding with minor oil resources: 2.8% (64.2 Gbbl) of conventional oil is located in the world's tropical rainforests, overlapping with 33% of the total area. 40% of the Amazon basin overlaps with oil resources.

Considering the carbon budget associated with the target of 1.5 °C of global warming and applying a cost-optimal distribution of the carbon budget among coal, gas, and oil (following Welsby et al.[8]), 752 Gbbl (29 per cent of 2575 Gbbl) of the conventional oil resources can be extracted (See Supplementary Table 1). We adopt this estimate of extractable oil and out of our 2276 Gbbl conventional oil resources we consider that 1524 Gbbl need to be left untapped (see the Methods section for a detailed explanation).

### Biological and social criteria
To identify these 1524 Gbbl of unburnable conventional oil resources, we use biological and social criteria. First, we define exclusion zones.

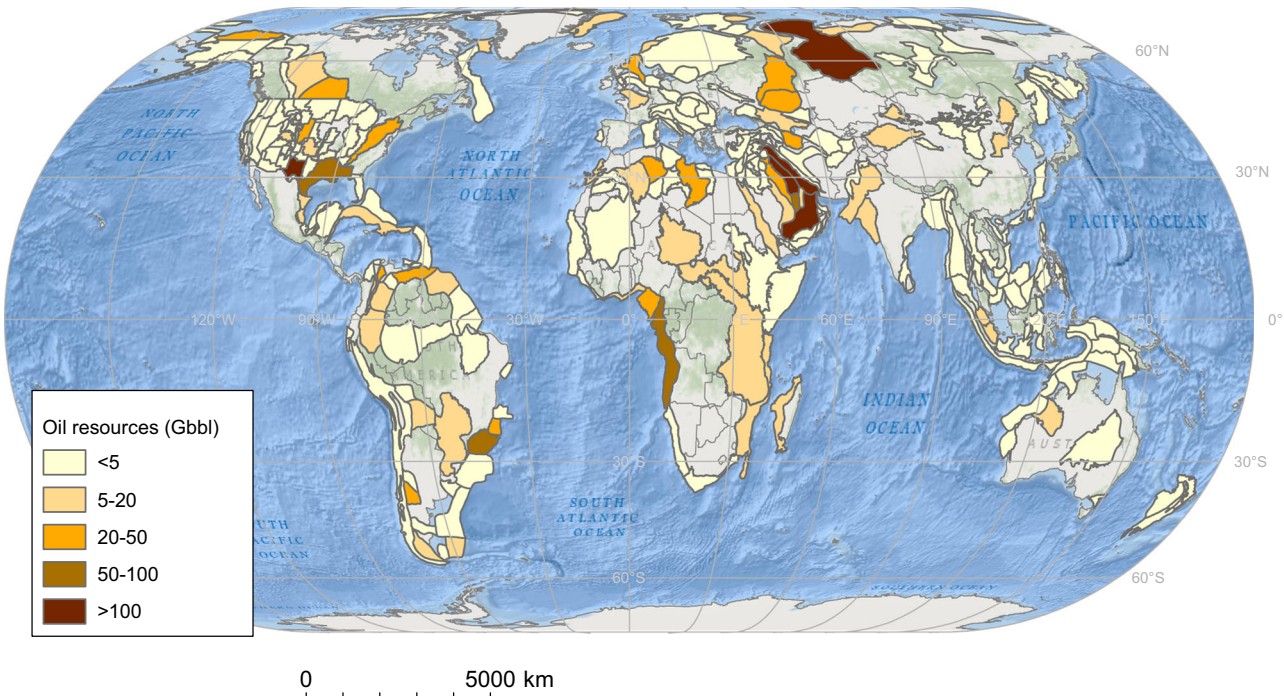

**Fig. 1 | Global distribution of conventional oil resources.** Volumes of conventional oil resources include technically recoverable conventional resources of oil, light tight oil (LTO) and natural gas liquids (NGL). Data on conventional oil resources is shown per sedimentary basins in gigabarrels of oil (Gbbl). Own elaboration based on data from the United States Geological Survey (USGS). Source data are provided as a Source Data file. Ocean basemap from Esri, Garmin, General Bathymetric Chart of the Oceans (GEBCO), National Geophysical Data Center from National Oceanic and Atmospheric Administration "(NOAA NGDC), and other contributors.

These are areas whose oil resources should be left untapped since they coincide with top-priority socio-environmental criteria. Second, we use complementary biological and social criteria to rank oil resources and create a priority list of unburnable resources in order to stay within the remaining carbon budget. We use several indicators to capture the biological value of specific geographical areas: biodiversity hotspots[63], richness centers of terrestrial and marine endemic species[64,65], and the global system of protected areas. The biodiversity hotspot approach[63] stands out as the best-known and the most widely accepted scheme to identify global biodiversity conservation priorities[66]. However, since biodiversity hotspots are circumscribed to terrestrial areas, we also considered the richness centers of terrestrial and marine endemic species in order to capture all global biodiversity conservation priorities[65]. Apart from biodiversity indicators, we have also included the global system of protected areas to align with existing conservation initiatives. The protected status of these areas reflects political and societal decisions and only partially coincide with global conservation priorities[64].

Regarding social criteria, considering the public health risks associated with oil extraction, we consider urban areas, with a 10 km buffer around them, irreconcilable with oil extraction[67,68]. Furthermore, we also use the presence of Indigenous Peoples in voluntary isolation as a criterion for unburnable conventional oil. People in voluntary isolation coinciding with oil resources exist in the Amazon, in Paraguay (Chaco), India (Andaman Islands), and Papua New Guinea (see Fig. 2).

The amount of conventional oil resources that overlap with biodiversity hotspots, terrestrial richness centers of endemic species, marine richness centers of endemic species, global natural protected areas, urban areas, and the territories of Indigenous Peoples living in voluntary isolation, is relatively small and account for 10.1% (231 Gbbl), 3.1% (71 Gbbl), 3.9% (89 Gbbl), 8.2% (186 Gbbl), 6.4% (146 Gbbl) and 0.1% (2.28 Gbbl), respectively, of the 2276 Gbbl of conventional oil

resources (Table 1). Altogether, conventional oil resources that overlap with top-priority socio-environmental characteristics only account for 26.8% (609 Gbbl) of global conventional oil resources. These 609 Gbbl are well below the 1524 Gbbl of our georeferenced conventional oil resources required to be left unburned to keep global warming under 1.5 °C.

These top-priority unburnable oil resources cover exclusion zones totaling 29.5 million km². They coincide with terrestrial biodiversity hotspots (11.2 million km², in particular those of the Caribbean islands, Tropical Andes, Cerrado, Atlantic Forest, Horn of Africa, Afromontane, and the North American Coastal Plain, and, important sections of the Indo-Burma, Sundaland, Irano-Anatolian and Caucasus biodiversity hotspots) and marine richness centers of endemic species (7.4 million km², in particular those of South-East Asia including the North Western coast of Australia, the Red Sea and the Caribbean) (see Fig. 2 and Table 1). The unburnable oil resources overlapping with urban (3.4 million km²) and natural protected areas (9.8 million km²) are distributed across continents. Most of these exclusion zones contain minor quantities of oil (Supplementary Table 3), and 60% of the identified top-priority unburnable conventional oil resources are located in small portions (2.8 million km²) of sedimentary basins from the Arabian Peninsula (Mesopotamian Foredeep Basin, 10%; Rub Al Khali Basin, 3%; Greater Ghawar Uplift, 2%), the Iranian Zagros mountains (Zagros Fold Belt, 12%), the US (Gulf Coast Basins, 10%; Appalachian Basin, 2%; Northern Alaska, 2%; Permian Basin, 1%), Venezuela (Maracaibo Basin, 4%, and East Venezuela Basin, 1%), Siberia (West Siberian basin, 4%), Mexico (Villahermosa Uplift, 2%), the Gulf of Guinea (Niger Delta, 2%), Madagascar (Morondava, 2%), the Red Sea (2%) and the Atlantic Forests (Santos Basin, 1%). Thus, while the Middle East is the region with more top-priority unburnable conventional oil in absolute terms (168.5 Gbbl), the region of Developing Asia (outside of China and India) stands out with approximately 78% of its resources considered unburnable (Table 2).

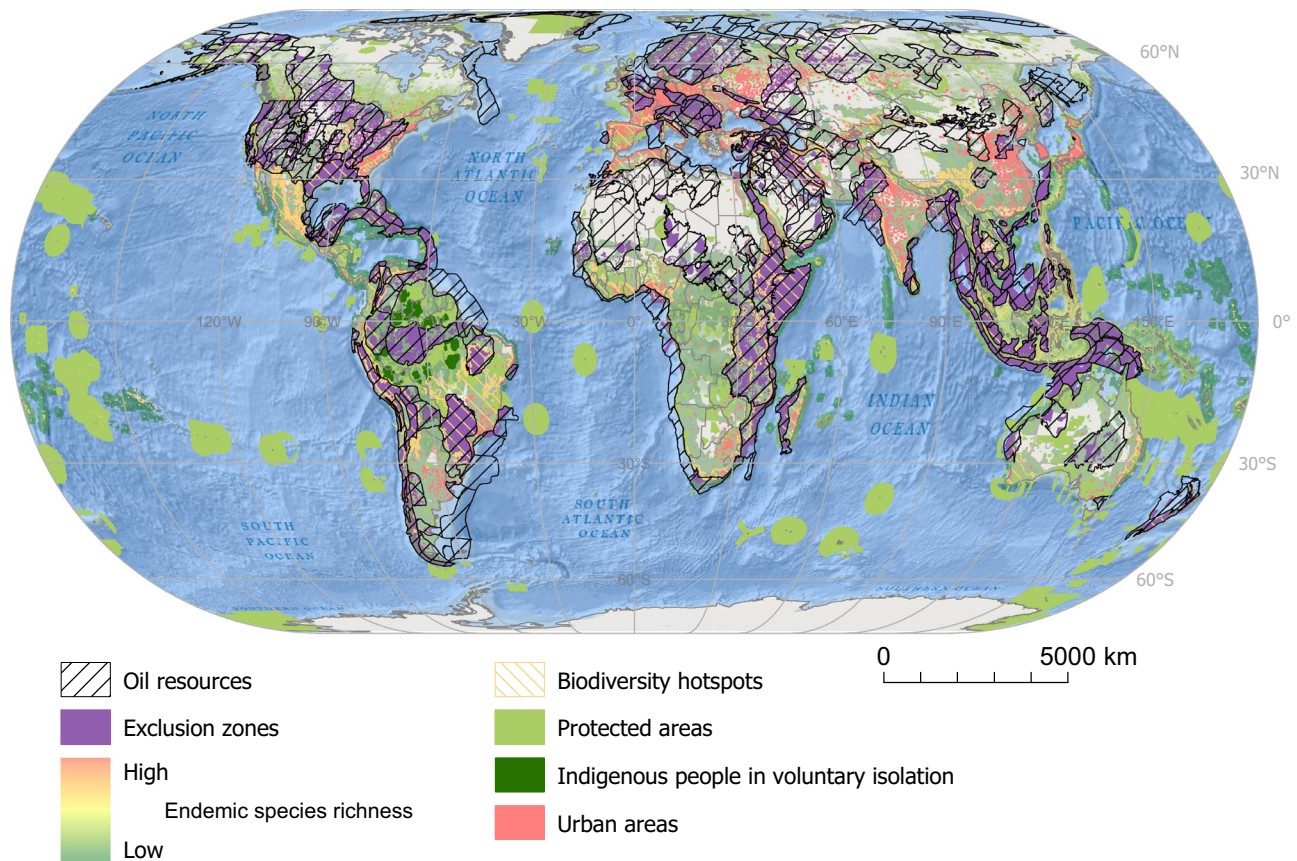

**Fig. 2 | Global distribution of top-priority unburnable conventional oil resources according to their coincidence with areas of outstanding socio-environmental characteristics.** Those are areas whose oil resources should be left untapped since they coincide with top-priority socio-environmental criteria are defined as 'exclusion zones'. Spatial data on conventional oil resources are from the United States Geological Survey (USGS); data on biodiversity hotspots from the Critical Ecosystem Partnership Fund (CEPF, Version 2016.1)[90]; data on richness of endemic species, from Jenkins et al.[93] and Jenkins et al.[94]; data on terrestrial and marine protected areas, from the World Database on Protected Areas (WDPA)[95]; data on urban areas, from the University of Wisconsin-Madison, Boston University and the MODIS Land Group[97,98]; and data on Indigenous Peoples in voluntary isolation from the Amazon Georeferenced Socio-Environmental Information Network (RAISG, in its Spanish acronym)[102] and Survival International[103]. Source data are provided as a Source Data file. Ocean basemap from Esri, Garmin, General Bathymetric Chart of the Oceans (GEBCO), National Geophysical Data Center from National Oceanic and Atmospheric Administration "(NOAA NGDC), and other contributors.

**Table 1 | Unburnable conventional oil resources and exclusion zones according to the different socio-environmental criteria**

| | | | Oil Resources Over-lapped (Gbbl) | % Global Conventional Oil Resources | Intersected surface (km2) |
|---|---|---|---|---|---|
| Biological criteria | Global Conservation Priorities | Biodiverstiy Hotspots (BH) | 230.8 | 10.1% | 11,196,748 |
| | Richness centers of endemic species (RCES) | Terrestrial | 71.3 | 3.1% | 4,352,413 |
| | | Marine | 89.0 | 3.9% | 7,364,686 |
| | | Total | 157.6 | 6.9% | 11,665,615 |
| | Protected Areas (PA) | | 185.8 | 8.2% | 9,777,282 |
| | Total | BH + PA | 397.9 | 17.5% | 19,324,082 |
| | | RCES + PA | 323.6 | 14.2% | 19,703,212 |
| Social criteria | Indigenous People in Voluntary Isolation (IPVI) | | 2.3 | 0.1% | 332,794 |
| | Urban areas (10 km buffer) - UA | | 145.7 | 6.4% | 3,443,184 |
| | UA + IPVI | | 148.0 | 6.5% | 3,775,978 |
| Total Exclusion zones | BH + PA + IPVI + UA | | 507.4 | 22.3% | 21,606,160 |
| | RCES + PA + IPVI + UA | | 454.5 | 20.0% | 22,833,044 |
| | Total | | 608.8 | 26.8% | 29,540,906 |

Conventional oil volumes located in the different exclusion zones are expressed in gigabarrels (Gbbl). Overlaps between conventional oil resources and biological and social criteria were calculated with QGIS Desktop 3.22.5 Białowieża with GRASS 8.2.1, ArcGIS Desktop 10.6 and ArcGIS Pro 3.0.3.

Apart from oil resources in the exclusion zones, additional resources should remain untapped to achieve the target of 1524 Gbbl of unburnable conventional oil. We have used continuous spatial data

## Table 2 | Regional distribution of unburnable conventional oil resources in exclusions zones

| Region | Conventional oil resources (Gbbl) | Unburnable conventional oil resources | |
|---|---|---|---|
| | | Gbbl | % |
| Africa | 298.6 | 83.3 | 27.9% |
| Australia and other OECD Pacific | 20.4 | 7.7 | 37.5% |
| Canada | 23.4 | 0.6 | 2.4% |
| China and India | 88.2 | 22.2 | 25.2% |
| Russia and former Soviet states | 343.5 | 45.5 | 13.3% |
| Central and South America | 319.0 | 117.8 | 36.9% |
| Europe | 77.7 | 12.9 | 16.6% |
| Middle East | 648.1 | 168.5 | 26.0% |
| Other Developing Asia | 54.9 | 42.6 | 77.6% |
| USA | 401.9 | 107.7 | 26.8% |
| Global | 2275.8 | 608.8 | 26.8% |

Conventional oil resources (total and unburnable) in each region are expressed in gigabarrels (Gbbl). See Supplementary Table 2 for the countries included in each region.

on the richness of terrestrial and marine endemic species and data on rural human population densities to identify the additional 915 Gbbl of unburnable oil resources needed. We constructed three scenarios, the first one based on human population densities (a social criterion) and the second and third ones based on the richness of terrestrial and marine endemic species (biological criteria) (see Figs. 3, 4 and 5). For example, taking rural population density as the criterion to prioritize unburnable oil resources, in addition to those in the exclusion zones, all oil resources coinciding with population densities above 1 person/ sq km would be unburnable.

## Discussion

In conclusion, we produced an atlas of unburnable oil resources whose operationalization would generate substantial collateral socio-environmental benefits. To this end, we first defined, based on biological and social criteria, exclusion zones strictly off-limits to oil activities. These zones overlap with 609 Gbbl of oil resources, while keeping global warming under 1.5 °C requires that 1524 Gbbl stay in the ground. Thus, oil resources in exclusion zones can be kept entirely untapped and still additional oil resources will also need to be left under the soil. Second, we created rankings to achieve the conservation of additional oil resources to reach a total of 1524 Gbbl of unburnable oil. The ranking prioritizes resources according to terrestrial and marine biodiversity and human population densities.

Our method to identify unburnable oil resources can be used flexibly and could be based on alternative or additional socio-environmental criteria—such as the presence of Indigenous Peoples, or the likelihood that exploitation would generate environmental

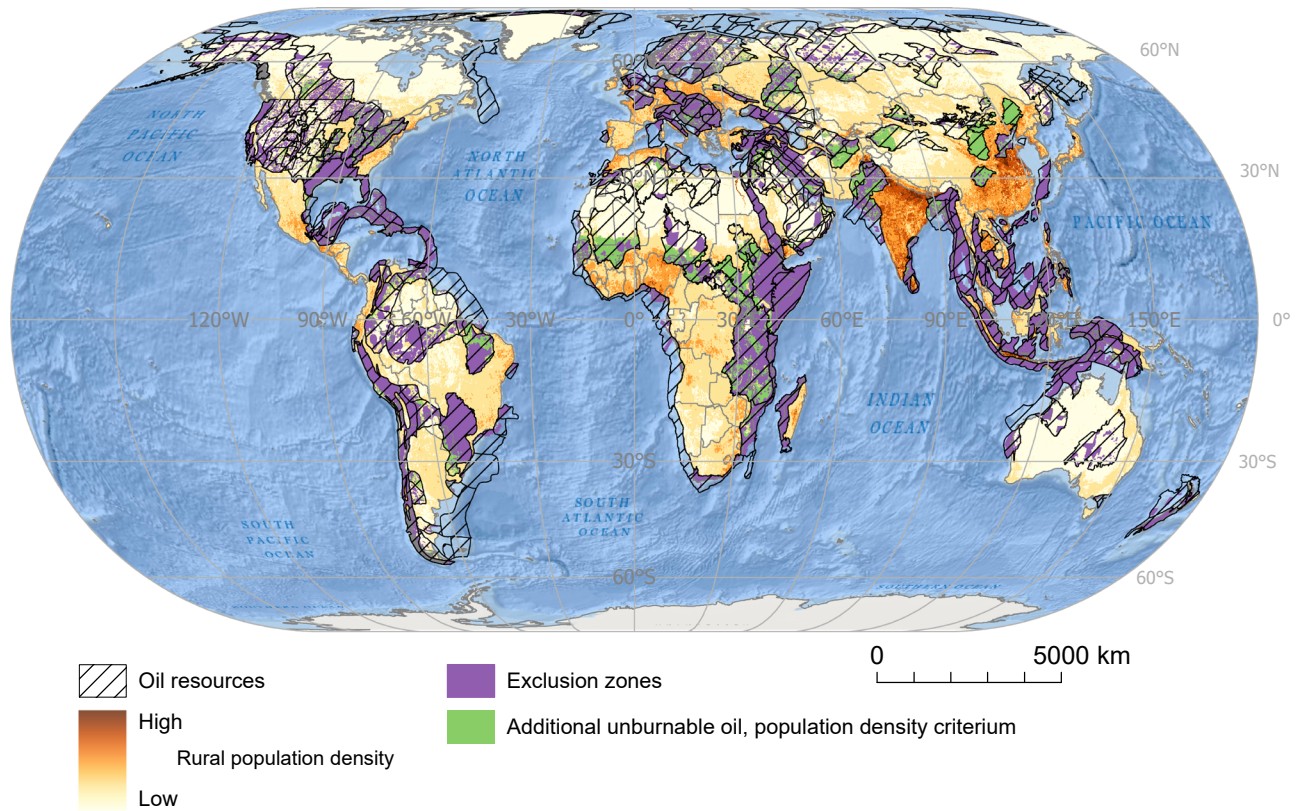

**Fig. 3 | Global distribution of additional unburnable conventional oil resources according to the social criterion.** Additional unburnable conventional oil resources are those conventional oil resources beyond the ones located in top-priority socio-environmental areas (i.e., exclusions zones) that should remain untapped to achieve the 1.5 °C target. Additional unburnable conventional oil resources are defined based on the density of human rural population. Spatial data on oil resources was retrieved from the United States Geological Survey (USGS); data on rural population density, from the Food and Agricultural Organization (FAO)[99]. Source data are provided as a Source Data file. Ocean basemap from Esri, Garmin, General Bathymetric Chart of the Oceans (GEBCO), National Geophysical Data Center from National Oceanic and Atmospheric Administration "(NOAA NGDC), and other contributors.

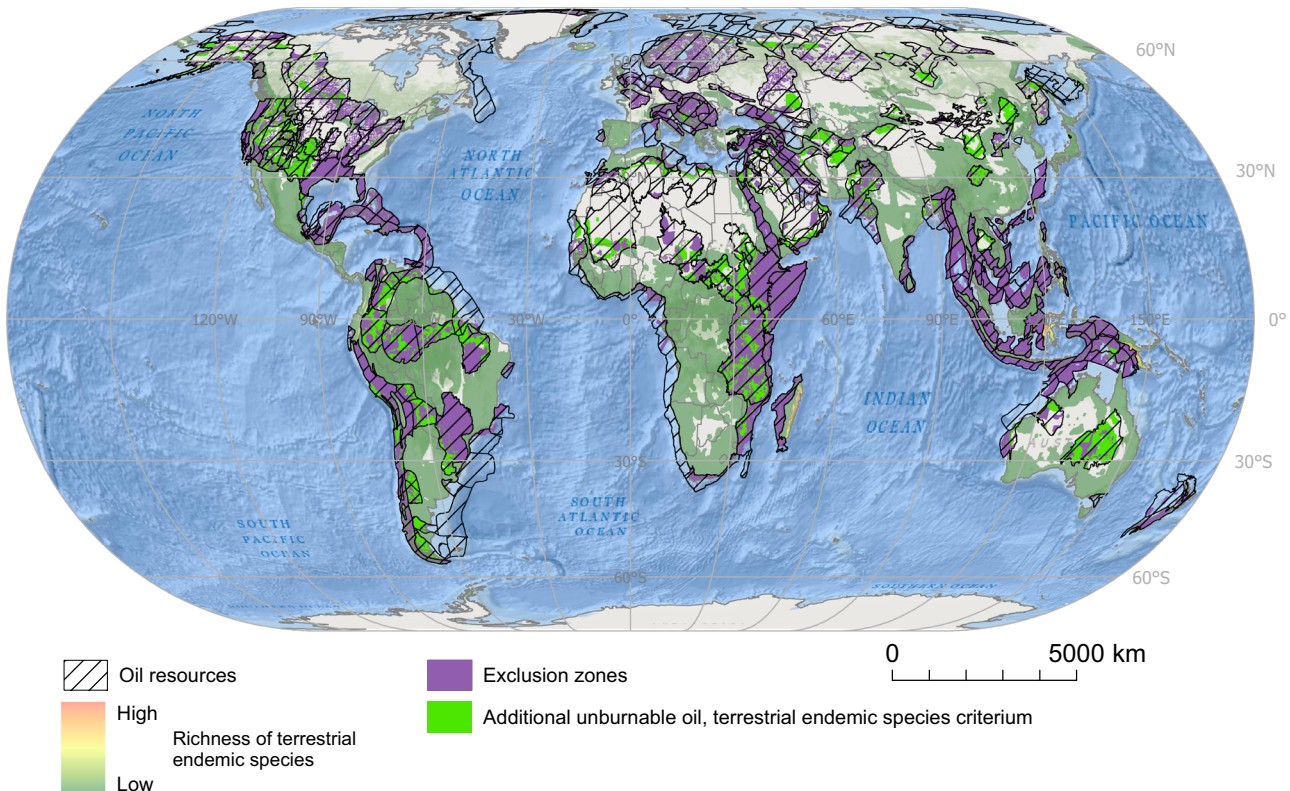

**Fig. 4 | Global distribution of exclusion zones and additional unburnable conventional oil resources according to the terrestrial biodiversity criterion.** Additional unburnable conventional oil resources are those conventional oil resources beyond the ones located in top-priority socio-environmental areas (i.e., exclusions zones) that should remain untapped to achieve the 1.5 °C target. Additional unburnable conventional oil resources are defined based on the richness of terrestrial endemic species. Spatial data on oil resources was retrieved from the United States Geological Survey (USGS); data on richness of terrestrial endemic species of mammals, amphibians[93]. Source data are provided as a Source Data file. Ocean basemap from Esri, Garmin, General Bathymetric Chart of the Oceans (GEBCO), National Geophysical Data Center from National Oceanic and Atmospheric Administration "(NOAA NGDC), and other contributors.

conflicts. Oil extraction in indigenous territories has been often associated with direct negative socio-environmental consequences and health risks[51,68,69]. Indigenous people have quite often opposed oil extraction in their territories[57] and their right to Free, Prior and Informed Consent, established by the United Nations Declaration on the Rights of Indigenous Peoples, has not been secured in most of the oil projects that overlap with indigenous territories[70]. The absence of consent could be used to demarcate unextractable oil resources. While there are no global data on this criterion available at this moment, this could be an important area for future research. Similarly, atlases of gas and coal could be produced, and specific risks associated with extraction technologies in certain contexts (such as resources in the Arctic and ultra-deepwater resources) could also be used as exclusion criteria. Also, the use of more fine-grained data on reserves and specific reservoirs and oil fields is key to producing atlases of unburnable oil that would be appropriate for local and regional policymaking. Additional research could also introduce ways to negotiate tradeoffs between different criteria. For example, future research could combine techno-economic factors alongside socio-environmental criteria to identify additional unburnable resources and include oil prices and production costs in the analysis[32]. This is particularly salient, since we are using data on 'technically recoverable resources' (as opposed to 'economically recoverable resources') and this category includes all the oil that can be produced based on current technology, industry practice, and geologic knowledge, regardless of their economic feasibility. Oil prices and production costs specific to each basin could be included in the analysis and multiple techno-economic factors combined alongside socio-environmental criteria to identify unburnable resources[31].

The atlas of unburnable fuels provides a simple guideline for operationalizing supply-side initiatives to complement the existing climate policy framework that focuses overwhelmingly on the demand for fossil fuels. It also makes it possible for energy corporations, governments and, more generally, investors to minimize the risks of stranded assets by highlighting those fossil fuel resources that overlap with socio-environmentally sensitive areas and, therefore, reduce the possibility of being impacted by future environmental policies (including and beyond climate policies) or becoming the target of contentious actions by social movements—e.g., divestment campaigns[71]. The case for declaring a substantial portion of oil resources unburnable is very strong if the increase in global average temperature is to be limited to 1.5 °C. The use of socio-environmental criteria to select unburnable resources can enhance the collateral benefits of climate policies.

## Methods
### Secondary data collection
Our work is based on the spatial analysis of global datasets on conventional oil resources, biodiversity hotspots, richness of endemic species, world protected areas, human population density, urban areas and territories of Indigenous Peoples in voluntary isolation. These datasets were compiled from governmental agencies, multilateral institutions, non-governmental organizations and standard-setting organizations. We used spatial and tabular data from the United States Geological Survey (USGS), the Food and Agriculture Organization of the United Nations (FAO), the United Nations Environment Program World Conservation Monitoring Center (UNEP-WCMC), the Critical Ecosystem Partnership Fund (CEPF), Conservation

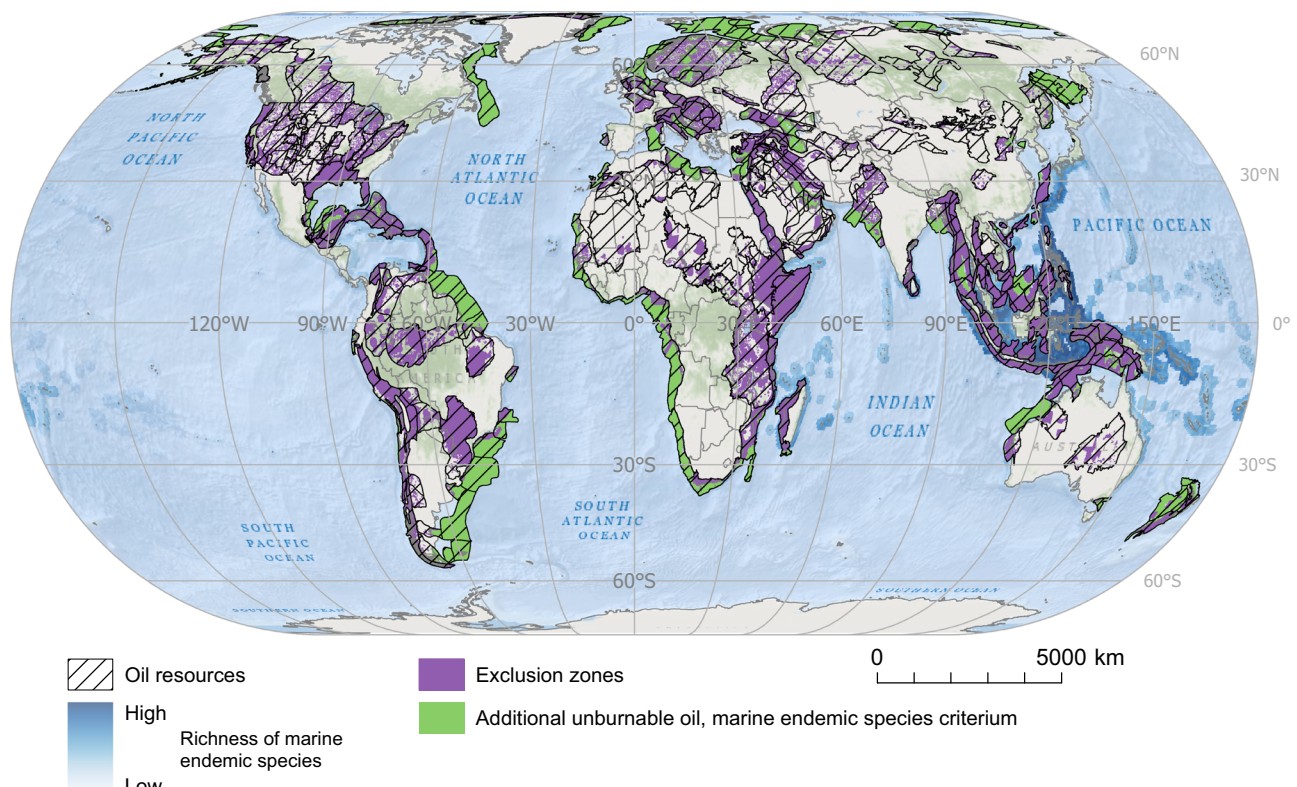

**Fig. 5 | Global distribution of exclusion zones and additional unburnable conventional oil resources according to the marine biodiversity criterion.** Additional unburnable conventional oil resources are those conventional oil resources beyond the ones located in top-priority socio-environmental areas (i.e., exclusions zones) that should remain untapped to achieve the 1.5 °C target. Additional unburnable conventional oil resources are defined based on the richness of marine endemic species. Spatial data on oil resources was retrieved from the

United States Geological Survey (USGS); data on richness of marine endemic species of plants, fish, echinoderms, crustaceans, cnidarians, mollusks, mammals, reptiles, and birds, from ref. 94. Source data are provided as a Source Data file. Ocean basemap from Esri, Garmin, General Bathymetric Chart of the Oceans (GEBCO), National Geophysical Data Center from National Oceanic and Atmospheric Administration "(NOAA NGDC), and other contributors.

International, Survival International and scientific literature on richness of endemic species. Details on data collection and management are provided below.

**Georeferenced datasets on oil reserves and resources.** The two most common metrics to report the availability of oil are 'oil reserves' and 'oil resources'. Oil reserves are the quantity of remaining oil that is recoverable under current economic conditions. Oil resources (or, remaining ultimately recoverable oil resources) denote the quantity of remaining oil that is recoverable over time with both current and future technology, irrespective of current economic conditions[72]. Thus, reserves are a subset of resources. Furthermore, oil is usually also categorized as 'conventional' and 'unconventional'. Here, following Welsby et al.[8], we define oil with density lower than water (often standardized as '10° API') as conventional (i.e., oil, LTO, and NGL) and the rest as unconventional (i.e., natural bitumen, extra-heavy oil, and kerogen oil). However, it is important to note that there is no full consensus on whether LTO should be considered conventional or unconventional oil and some of the world energy standard setting institutions (i.e., USGS, SPE, EIA and IEA) define LTO as unconventional[59–62]. According to these institutions, unconventional oil lacks the porosity and permeability of conventional reservoirs required to flow without stimulation and require specialized extraction technology (e.g., hydraulic fracturing stimulation for LTO).

Although reserves are more likely to be extracted than resources, our analysis is focused on resources for several reasons. From a policy perspective, an analysis of resources is a first logical step towards the fulfillment of the Paris Agreement commitments since that can guide

investment in oil exploration. At the same time, reserve volume estimates fluctuate over time depending on prices, the cost of available technologies, the development of new oil extraction technologies, new discoveries, and strategic overestimation by rights holders[73,74]. There is much uncertainty about future energy prices which heavily depend on some key political choices, including the climate mitigation policies adopted by governments around the world. Ultimately, reserves depend crucially on climate policies, turning an analysis focusing on them into a questionable policy tool. Nevertheless, we acknowledge that an analysis of reserves is an important next step in the research on unextractable fossil fuels and our methodology could be adapted to construct atlases for unburnable fossil fuel reserves.

Using a remaining carbon budget of 580 GtCO$_2$, Welsby et al.[8] established that 744 Gbbl (58%) of current oil reserves, both conventional and unconventional, should remain unburned, while 1823 Gbbl (71%) and 1513 Gbbl (99%) of conventional and unconventional oil resources, respectively, were considered unburnable. This distribution of unburnable oil categories was based on the production costs of different resources, taking into account extraction, refining, and transport costs. Thus, according to Welsby et al.[8], the overwhelming majority of unconventional oil should remain unburned because its production was considerably less economic than the production of other oil categories. Building on Welsby et al.[8], our study focuses on the identification of those conventional oil resources that should remain unburned according to social and biological criteria.

For that purpose, we compiled a dataset of global conventional oil resources based on data from the United States Geological Survey (USGS)[41]. For world conventional oil resources, we used data from the

World Oil and Gas Assessments produced by the USGS World Energy Project, in particular the USGS World Petroleum Assessment (WPA) 2000[75] and 2012[76] and, 27 regional USGS assessment reports on undiscovered oil and gas resources in priority geologic provinces in the World published between 2012 and September 2022[77]. The USGS WPA 2000 provides estimates of the quantities of remaining technically recoverable conventional oil resources outside the United States that had the potential to be added to reserves from 1995 to 2025 (i.e., discovered reserves + contingent resources, according to the definition of SPE[62]). We used spatial and tabular data on remaining oil and remaining NGL (i.e., quantities of conventional oil and NGL excluding reported cumulative volume of oil and NGL that had been already produced) from the WPA 2000 ("wep_prvg.shp" geospatial data set). From the USGS WPA 2012, we have retrieved estimates of the quantities of undiscovered (or 'prospective' according to the definition of SPE[62]) conventional oil resources. Specifically, we used spatial and tabular data on undiscovered technically recoverable oil resources (i.e., the oil that could be produced using available technology and industry practices regardless of economic or accessibility considerations) from the WPA 2012 ("Province Summary.xls").

When individual regional reports for specific sedimentary basins have been issued after 2012, undiscovered conventional oil volumes from the WPA 2012 have been updated. Thus, conventional oil resources volumes have been calculated by adding remaining discovered conventional oil resources as in the WPA 2000, undiscovered conventional oil resources as in the WPA 2012, and individual USGS assessments for sedimentary basins from 2012 onwards. For USA conventional oil resources, we used data from National Oil and Gas Assessments (NOGA) produced by the USGS. We used tabular data on conventional oil resources volumes for USA geological provinces from the USGS NOGA Resources Update[78] and 6 USGS national assessment reports on undiscovered oil and gas resources published between 2013 and September 2022[79]. The spatial data for USA geological provinces were acquired from the USGS NOGA 1995[80] and the NOGA Province Boundaries update from 2012[81]. These are the most accurate and up to date open-access and available georeferenced datasets on conventional oil resources at the global level. However, it is important to note that these estimates of the global conventional oil resources are subject to some degree of uncertainty (as demonstrated by the variation of the estimates over time and across different sources)[82]. One limitation of our data is that we do not account for (a) conventional oil extraction between 2000 and 2022 (2013–2022 for USA) and, (b) discoveries made between 2000 and 2012+ (i.e., resources added to contingent/discovered resources from prospective/undiscovered resources –according to the definition of SPE- between the WPA 2000 and the WPA 2012 or the individual USGS assessments published for each particular sedimentary basin from 2012 onwards). Although there is no available data on the conventional oil extracted in each basin between 2000 and 2022, 751 Gbbl of conventional oil have been extracted globally in this period[83]–see Supplementary Table 4 in Supplementary Information. While this omission tends to overestimate the amount existing resources, the additions to the contingent resources category from the prospective resources category between 2000 and 2012+ produces an underestimation. It is worth mentioning that between 2000 and 2022, global reserves have increased by 685 Gbbl (585 Gbbl between 2000 and 2012)[84]. Future research, when updated data on oil extraction and discovered resources become available at the basin level, could gauge the implications of updated oil resource estimates for the spatial distribution of unburnable oil.

There are significant differences between the georeferenced resources we use, and the non-georeferenced tabular data provided by other sources and datasets. However, we believe the two are close enough in both definition and size to be a meaningful proxy. We put together georeferenced data for 2276 Gbbl of conventional oil resources, while Welsby et al.[8] provided a global estimate of 2575 Gbbl.

This difference is explained by the dearth of accessible and up-to-date spatial data. Welsby et al.[8] estimated that, based on the remaining carbon budget, 752 Gbbl of conventional oil resources could be burned. We subtracted this amount from our own resource estimates to determine the amount of conventional oil resources that cannot not be extracted. Thus, out of the 2276 Gbbl of georeferenced conventional oil resources, 1524 Gbbl (2276 minus 752 Gbbl) should be left untapped and we use socio-environmental criteria to rank and prioritize these resources. If any of the other conventional oil resources not considered in this study (because of the unavailability of georeferenced data) are burned, a larger portion of the conventional oil resources for which we have georeferenced data should be left untapped in order to limit global warming to 1.5 °C. The global energy systems model used by ref. 8 assumed a 2018–2100 carbon budget of 580 $GtCO_2$.

To identify the unburnable oil resources whose conservation would generate substantial collateral socio-environmental benefits, our spatial analysis builds upon the cost-optimal distribution of unburnable fuels among coal, gas and oil proposed by ref. 8 (i.e., a key assumption of our spatial analysis is that, as set by Welsby ref. 8, 71% of the conventional oil resources should remain unextracted by 2050, as well as 81% of conventional gas resources, 99% of unconventional oil resources, 93% of unconventional gas resources and 97% of coal).

The equivalences used to convert different types of fossil fuels into $CO_2$ emission are the following: 0.43 metric tons $CO_2$/barrel of oil equivalent; 0.0548 metric tons $CO_2$/Mcf (thousand cubic feet) of natural gas; $9.05 \times 10^{-4}$ metric tons $CO_2$/pound of coal[85,86].

Regarding our spatial unit of analysis, we have used the 313 Assessment Units of the WPA 2012[76], located in one of the 171 geologic provinces of the world, and 67 USA geological provinces[78]. Geological provinces are USGS-defined areas having characteristic dimensions of hundreds to thousands of square kilometers encompassing a natural geologic entity (i.e., sedimentary basin, thrust belt, delta), or some combination of contiguous geologic entities. We are aware that the use of finer spatial units of analysis, such as oil fields or reservoirs, would be a significant methodological improvement, because it would bring mapping to scales comparable with regional decisions on which oil fields and reservoirs should be kept untapped. However, these data are inaccessible for scientific purposes, or are available at a substantial price from private sources. Access to complementary spatial datasets on gas and coal resources, but also on other up-to-date spatial data on conventional oil categories, is needed to provide a complete overview for policy makers.

**Georeferenced datasets for biological criteria.** Different datasets have been used to identify oil resources that overlap global biodiversity conservation priorities[66]. We have used two different schemes: biodiversity hotspots[63] and centers of diversity for endemic terrestrial and marine species[64,65]. To align our proposal for unburnable conventional oil resources with current efforts and measures for biodiversity protection worldwide, we have used a spatial database of the natural protected areas[87]. Furthermore, to rank additional oil resources beyond the top-priority unburnable ones, we have also used global data on the geographic distribution of terrestrial and marine endemic species[64,65].

Biodiversity Hotspots is the best-known and the most widely accepted Global Biodiversity Conservation Priorities scheme for terrestrial ecosystems. Biodiversity Hotspots were defined by Norman Myers[63,88,89] and reassessed up to the current 36 biodiversity hotspots designated in 2016. To qualify as a hotspot, a region has to contain at least 1500 species of vascular plants as endemics and have 30 percent or less of its original vegetation remaining. Among conservation biologists there is broad consensus on the use of the Biodiversity Hotspot approach and it has been adopted by standard-setting organization worldwide such as Conservation International, the Critical Ecosystem

Partnership Fund (CEPF), and the Global Environment Facility (GEF). We retrieved the georeferenced data from CEPF (Version 2016.1. 25 April 2016)[90].

Although existing global conservation priorities are based on biodiversity hotspots, there are also other approaches. Thus, making use of species-distribution databases, such as the Global Amphibian Assessment or the Global Mammal Assessment[91,92], Jenkins et al. reassessed terrestrial[64] and marine[65] conservation priority areas. Using new data on >21,000 species of mammals, amphibians, and birds, and focusing on endemic species, Jenkins et al. identified conservation priority areas for vertebrates[64]. Similarly, Jenkins et al.[65] assessed global marine biodiversity conservation priority areas by evaluating the geographic ranges of 4352 marine species of 9 different taxa: plants, fish, echinoderms, crustaceans, cnidarians, mollusks, mammals, reptiles, and birds[65]. Jenkins's terrestrial and marine priority areas do not only reflect today's improved knowledge of biodiversity, but also consider both marine and terrestrial ecosystems. On the contrary, (plant-based) Biodiversity Hotspots are circumscribed to terrestrial areas. Therefore, we also used these centers of diversity for endemic terrestrial and marine species as an alternative scheme of global biodiversity conservation priorities to identify the unburnable conventional oil reserves and resources.

GIS data on biodiversity conservation priority areas for terrestrial vertebrates and on the distribution of >21,000 species of mammals, amphibians, and birds were retrieved from the Conservation Science Around the World Website[93] (accessed January 2019). GIS data on geographic ranges of marine species were retrieved from the Dryad Digital Repository[94].

The dataset we used for the planet's protected area system was the *World Database on Protected Areas* (WDPA)[87]. The WDPA is a joint project between the United Nations Environment Program (UNEP) and the International Union for Conservation of Nature (IUCN). It is compiled and managed by the UNEP World Conservation Monitoring Center (UNEP-WCMC). The WDPA is the most up to date and comprehensive global database of marine and terrestrial protected areas, updated monthly in collaboration with governments, non-governmental organizations, academia and industry. It is made available online through Protected Planet[95]. The WDPA version we used was updated in July 2020. The WDPA includes all six IUCN Protected Area Management Categories, from strictly protected IUCN categories (I–IV) to the lowest protection level (V-VI), and protected areas for which there is no IUCN Protected Area Management Category.

**Georeferenced datasets for social criteria.** To assess the social value of conventional oil resources we used the following datasets: urban areas, territories of Indigenous Peoples in voluntary isolation, and rural population density.

Regarding data on urban areas, we used the MODIS 500-m global map of urban extent produced by the University of Wisconsin-Madison, Boston University and the MODIS Land Group[96]. 'Urban areas' are identified through remote sensing based on physical attributes (not on population densities). The 'MODIS 500-m global map of urban extent' defines 'urban areas' as pixels that are dominated by the built environment. 'Built environment' includes all non-vegetative, human-constructed land covers, such as buildings, roads, runways, etc., and 'dominated' implies coverage greater than or equal to 50% of the pixel[97,98]. All these areas, including a 10 km buffer around them, were designated as irreconcilable with oil extraction. Although these 'urban areas' also include industrial areas (i.e., areas that could arguably be suitable for oil extraction), they are, to our knowledge, the best proxy for global data on densely populated areas (both rural and urban). The 10 km buffer was set considering health risks associated with oil extraction (see 'Supplementary Information' for a detailed explanation of the 10 km safe distance).

For the areas beyond the 'built environment', we used a dataset for rural population density: the rural population density map of the Food Insecurity, Poverty and Environment Global GIS Database (FGGD)[99]. FGGD is a global database maintained by the Food and Agricultural Organization (FAO) to analyse food insecurity and poverty in relation to the environment. The FGGD rural population density map provides estimates of the global population distribution in 2015. It is a global raster datalayer with a resolution of 5 arc-min. Each pixel classified as non-built environment by the urban area boundaries map contains the number of persons per square kilometer, aggregated from the 30 arc-s datalayer. The method used by FAO to generate this datalayer is described in ref. 100.

We also used the presence of Indigenous Peoples in voluntary isolation as a criterion for primary selection of unburnable conventional oil resources. Indigenous Peoples in voluntary isolation do not maintain sustained contact with non-indigenous populations, and generally avoid it because most of the previous contacts have been violent and have had serious consequences for them[101]. Given their situation of isolation with respect to the non-indigenous societies, they do not have the immunological defenses against relatively common diseases, and contagions often have devastating effects on them causing outbreaks with high rates of mortality. Groups in voluntary isolation or initial contact exist in the Amazon as well as Paraguay (Chaco), India (Andaman islands), and Papua New Guinea. The dataset used for territories of Indigenous Peoples in voluntary isolation was pieced together from two different institutions: the Amazon Georeferenced Socio-Environmental Information Network (RAISG, in its Spanish acronym) and Survival International. RAISG is a consortium of civil society organizations from the Amazon countries that produces comprehensive socio-environmental geospatial data on Amazonia[40]. GIS data from RAISG[102], was used for the territories of Indigenous peoples living in voluntary isolation in the Amazon. GIS data from Survival International[103] was used for the territories of Indigenous peoples living in voluntary isolation in the South-East Asia. Survival International, founded in 1969, is one of the key civil society organizations defending the rights of Indigenous Peoples.

**Spatial analysis**

Spatial analyses were conducted using Geographical Information Systems (QGIS Desktop 3.22.5 Białowieża with GRASS 8.2.1, ArcGIS Desktop 10.6 and ArcGIS Pro 3.0.3.) to calculate overlaps between different land-use categories in order to select the conventional oil resources to be kept off-limits of oil extraction. Two different workflows were used to pinpoint priority unburnable conventional oil resources whose conservation would generate substantial collateral socio-environmental benefits: exclusion areas and conditional areas. We first identified 'exclusion areas', or portions of geologic provinces that should be set as strictly off-limits to oil extraction because they overlap critical areas for global biodiversity conservation or because of likely human rights implications of oil extraction. Conventional oil resources that coincide (i.e., 'intersect' tool in QGIS and ArcGIS) with areas that are highly significant for at least one of these two dimensions were classified as strictly unburnable. These top-priority unburnable oil resources were those located in areas considered irreconcilable with fossil fuel extraction where operations should be avoided in any circumstance. To do so, we applied a spatial suitability analysis based on Boolean overlay[104]. Thus, to qualify as an 'exclusion area', a geological province, or part of it, should meet one or more of the following criteria (for biodiversity exclusion areas): conventional oil resources are overlapped by centers of diversity for endemic vertebrates (as defined by ref. 64) or by global conservation marine biodiversity priorities (as defined by Jenkins and Van Houtan[65]) or by one of the 36 biodiversity hotspots (as defined by Norman Myers and reassessed in 2016[63]) or by Natural Protected Areas (as defined at the WDPA[87]);

(for social exclusion areas) conventional oil resources are overlapped by urban areas[96] (as defined by Schneider et al.[97]) with a 10 km buffer around them or by territories of Indigenous peoples living in voluntary isolation (as defined by RAISG[40] and Survival International[103]).

Second, we identified conditional areas to be set aside from oil production. Since unburnable conventional oil resources located in exclusion areas are not enough to limit cumulative carbon emissions in line with a carbon budget associated with 1.5 °C warming, supplementary conventional oil resources need to remain unburned. In fact, exclusion areas just account for 609 Gbbl of conventional oil resources, while an extra 915 Gbbl need to be left untapped to reach the total target of 1524 Gbbl of unburnable conventional oil. We developed three different scenarios of potential supplementary resources to remain unused: one exclusively based on a social criterion, and two solely based on biological criteria. In the first scenario, resources were designated as unburnable based on human rural population density. Up to 915 Gbbl, those resources that overlap more densely populated rural areas, were primarily selected. In the second and third scenarios, resources were classified as unburnable based on marine and terrestrial endemic species diversity, respectively. Again, up to 915 Gbbl, those resources with higher biodiversity for marine endemic species were firstly selected. The same method was used for terrestrial endemic species richness to construct the third scenario. The dataset on endemic terrestrial species diversity is based on data from 3 different taxa (i.e., mammals, amphibians, and birds), while for small-ranged marine species diversity, is based on 9 different taxa (i.e., plants, fish, echinoderms, crustaceans, cnidarians, mollusks, mammals, reptiles, and birds). To avoid bias due to the different number of taxa and different data distribution even after normalization, datasets on endemic marine and terrestrial biodiversity could not be combined.

For all criteria, both social and biological, geologic assessment units were selected fully or partially, according to their overlap with the criteria. Thus, we conducted a pixel-based analysis. To do so, the shapefiles of geological provinces were rasterized to the same grain as the raster datasets for each criterion. Conventional oil resources present at each geologic assessment unit were divided and proportionally assigned to each pixel of the geologic assessment unit according to its surface area. This is, in fact, the main limitation of the data used here. All datasets were projected using Eckert IV equal-area.

Our method to identify unburnable oil resources can be used flexibly and could be based on additional socio-environmental criteria –such as e.g., the presence of Indigenous Peoples, or the likelihood that exploitation would generate environmental conflicts.

### Reporting summary
Further information on research design is available in the Nature Portfolio Reporting Summary linked to this article.

## Data availability
We follow the FAIR principles (i.e., Findable, Accessible, Interoperable and Reusable data) and the principles of EU Open Science Policy. All the data generated in this study have been deposited in the Digital Repository of the corresponding author's university (University of Barcelona): https://doi.org/10.34810/data891. Source data are provided with this paper. The sources and hyperlinks to the raw data used in this study are provided in Source Data file. Only open-access databases/datasets have been used in the study. Data on conventional oil resources outside the United States were retrieved from: USGS World Petroleum Assessment 2000 [https://pubs.usgs.gov/dds/dds-060/]; Undiscovered Conventional Oil and Gas Resources of the World 2012 [https://pubs.usgs.gov/publication/ds69FF]; World Undiscovered Assesments (2012–2022) [https://www.usgs.gov/centers/central-energy-resources-science-center/science/world-oil-and-gas-resource-assessments#publications]. Data on conventional oil resources in the

United States were retrieved from: USGS NOGA Resources Update-United States Geological Service (USGS). USGS National Assessment of Oil and Gas resources (NOGA) update (2013). [https://certmapper.cr.usgs.gov/data/noga00/natl/tabular/2013/Summary_13_Final.xls]; USGS national assessment reports on undiscovered oil and gas resources (2013–2022) [https://www.usgs.gov/programs/energy-resources-program/science/science-topics/national-oil-and-gas-assessment]. Geospatial data on conventional oil basins outside the United States were retrieved from: World Petroleum Assessment 2000 [http://certmapper.cr.usgs.gov/data/wep/dds60/wep_prvg.zip]; Undiscovered Conventional Oil and Gas Resources of the World 2012 [https://pubs.usgs.gov/publication/ds69FF]. Geospatial data on conventional oil basins in the United States are retrieved from: National Assessment of United States Oil and Gas Resources 1995. Circular 1118. [https://pubs.usgs.gov/circ/1995/1118/report.pdf]. National Oil and Gas Assessment Province Boundaries 2012 [https://certmapper.cr.usgs.gov/data/noga00/natl/spatial/geodatabase/usprov12gdb.zip]. Geospatial data on *Biodiversity Hotspots* were retrieved from: Critical Ecosystem Partnership Fund (CEPF) Biodiversity Hotspots 2016 [https://www.arcgis.com/home/item.html?id=fb8ec2af7cfc40c7af89d9b7e922d4d8]. Geospatial data on Centers of diversity for endemic species were retrieved from: Jenkins et al.[64] for terrestrial endemic species [https://biodiversitymapping.org/]; Jenkins et al.[65] for marine endemic species. Geospatial data on World Protected Areas were retrieved from: UNEP-WCMC Protected Planet, World Protected Areas Database. [https://www.protectedplanet.net/en/thematic-areas/wdpa?tab=WDPA]. Geospatial data on Urban Areas were retrieved from: Center for Sustainability and the Global Environment, Global Maps of Urban Extent from Satellite Data [https://sage.nelson.wisc.edu/data-and-models/datasets/#urbanextent]. Geospatial data on Rural population density were retrieved from: FAO Rural population density 2000 [http://www.fao.org/geonetwork/srv/en/resources.get?id=14052&fname=Map_2_2.zip&access=private]. Geospatial data on Presence of Indigenous Peoples in Voluntary Isolation were retrieved from: RAISG [https://www.amazoniasocioambiental.org/en/maps/]; Survival International [http://www.uncontactedtribes.org/where]. Source data are provided with this paper.

## Code availability
Our work is based on spatial analyses that were conducted using open source and commercial Geographical Information Systems (GIS): QGIS Desktop 3.22.5 Białowieża with GRASS 8.2.1, ArcGIS Desktop 10.6 and ArcGIS Pro 3.0.3. We have not used any custom code or mathematical algorithm, but the conventional tools provided by the GIS software used.

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

## Acknowledgements
This work was supported by the Spanish *Ministerio de Ciencia e Innovación* through grants RYC-2016-21366 MCIU/AEI/FSE, UE (M.O.M.), RTI2018-095949-BI00 MCIU/AEI/FEDER, UE (M.O.M.) and TED2021-132007B-I00 funded by MCIN/AEI/10.13039/501100011033 and by the European Union NextGenerationEU/PRTR (M.O.M.).

## Author contributions
L.P., M.O.M., M.A. and C.M. conceptualized the research. M.O.M. designed the methodology, assembled the original spatial data, supervised data processing, and wrote the original draft. G.M. conducted all the spatial analysis and calculated the overlaps. G.R.T. collected global data on undiscovered conventional oil resources from 2012 onwards. L.P. substantively contributed to write and revised the original draft. L.P., M.O.M., M.A., G.M. and G.R.T. revised the paper.

## Competing interests
The authors declare no competing interests.
