## [Peer Review File · Nature Communications]

REVIEWER COMMENTS

Reviewer #1 (Remarks to the Author):

This is a really interesting paper that considers how to prioritise what conventional oil resources need to stay in the ground based on a range of social and environmental criteria, thereby creating an atlas of exclusion areas for extraction. This is a welcome addition to the literature that is often dominated by economic efficiency objectives in terms of unburnable reserves and resources.

However, there are three major concerns / issues I'd like to raise in relation to the current manuscript.

1. Climate ambition

The paper is framed in terms of 2°C at 50%. I understand you have chosen this as it fits with the McGlade and Ekins (M&E) paper, from which determine % unburnable resources. However, much of the current discourse is around the transition away from fossil fuels under 1.5°C. Furthermore, the problem with 2°C at 50% is that it is not consistent with the Paris Agreement (of 'well below 2°C').

The Welsby et al. 2021 paper could be considered as an alternative source, although perhaps you did not pick this up due to the timing of its publication? In that paper, unextractable estimates for conventional oil resources sit at 1823 Gb, or 71%. This is quite a lot higher than in M&E. These numbers, which are used in SI Fig 1, could be requested from the authors.

My suggestion is that this paper would have more impact if framed using stronger ambition and would be more relevant to the current debate around fossil fuel phase out.

One additional point that needs to be recognised in relation to budgets discussed in the paper; the budget quoted in M&E was for 2011-2050, while other budgets such as those from Matthews et al are from 2020-2100. You need to make that clear. Also, M&E didn't run their model with a carbon budget but used a climate module; they mention a budget in their abstract but do not report a budget derived from their own analysis. This also needs to be made clearer in your paper.

2. Unburnable conventional oil resources

I am little confused with the unburnable numbers that you are using. You start with total conventional oil resources of 1818 Gb but state that '611 Gbl (263 GtCO₂) out of 1,818 Gbl of conventional oil resources need to be left untapped'. How do you determine 611 Gb? This seems to assume a 33% unburnable level, which is the reserves percentage from M&E. However, I think you want to use 54%, for conventional oil resources, as per M&E, which would give you an unburnable value of 981 Gb. If I have misunderstood this, then please make your calculations and assumptions clearer, being more explicit about the numerator/denominator for these estimates.

If this does indeed need correcting, it means that you may need to re-think / redo your three additional scenarios, as the gap between that which is excluded (457 Gb) and total unburnable resources will be significantly larger.

Finally, given that you focus on resources, not reserves, it may be worth re-thinking the introduction that focuses on reserves and could cause confusion (lines 40-45). The focus of the paper is on conventional oil resources so perhaps worth focusing on this, in respect of this context setting section – and keeping any discussion of reserves in the Methods.

3. Regional distribution

I think this paper would be significantly strengthened by further elaborating on the distribution of oil resources by region and making some comment on how this differs from M&E or Welsby et al.

Firstly, it would be good if you were able to state the change in regional distribution of unburnable resources. Secondly, a comparison with the cost-optimal distributions from other papers would highlight which regions would be most impacted by alternative socio-environmental criteria – and reveal the tension discussed by Lenferna in his paper between efficiency and other criteria for prioritising unburnable oil.

Other briefer comments:

- Line 9-12. The way this is written sounds like M&E assumed 1370 MtCO₂ budget, which they didn't
- Line 17. The figure of 611 needs checking – and amending if necessary
- Line 31-35: for latest budget, better to refer to recent AR6 WG1
- Line 40-41: reference 6 says nothing about current reserves
- Lines 40-45: the discussion on reserves here could confuse the reader. The focus of the paper is on conventional resources so perhaps worth focusing on this, in respect of this context setting section.

- Line 49: the latest Production Gap report provides a useful listing of the various supply side efforts around the world.
- Lines 51-55: it would be worth mentioning the recent BOGA initiative that gained much attention at COP26
- Line 88-91: will you publish the datasets, including spatial layers? This would be important for allowing others to develop and use such information
- Line 216-217: implications also for donor and investment community, providing financing
- Line 280/296: M&E didn't use a carbon budget but rather a climate module to determine unburnable fossil fuels under a 2C target
- Line 282-283: You state 'This means that 611 Gbl (263 GtCO₂) of our 1,818Gbl (781 GtCO₂) of georeferenced conventional oil resources should be left untapped.' You need to provide information on how you get to 611 Gb (as per earlier comments)
- Line 291-295: how does the cost-optimal distribution inform your analysis? Can you elaborate on this? Also, do you start with the regional estimates of unburnable conventional oil resources from M&E for your analysis or simply take the global total, developing the distribution from that basis?

Reviewer #2 (Remarks to the Author):

This study makes a valuable contribution to debates about “unburnable carbon” and about co-benefits of decarbonisation.

It is well known that the world's fossil fuel reserves significantly exceed what can be burned while achieving the Paris climate goals, and therefore only a portion of reserves can be safely extracted. The key question is: which portion? This study answers that it should be the portion where environmental and social costs are lowest, and proposes a method for assessing oil provinces accordingly. By marrying spatial data on oil resources, on areas of biodiversity importance and on potential social impacts, the study shows how far environmental and social criteria can be used to shape decisions on which fossil fuels should remain unextracted.

However, I believe the study requires some further work before publication.

In particular, I think there is a methodological flaw, namely neglecting or mischaracterising the time dimension of oil production. McGlade & Ekins estimate an amount of oil that can be consumed between 2010 and 2050 in a 2°C scenario; the authors appear to interpret this wrongly as an amount that can be consumed from the present day until the ultimate end of oil production (e.g. the discussion of carbon budgets is effective from 2020). The geographical location of the oil resources is taken from USGS resource estimates from 1995, 2000 and 2012; these too are apparently taken to represent the present picture. In reality, reserves and resources change over time: today's reserves are equal to the reserves at prior date X, minus the amount extracted since X, plus the amount discovered since X, plus the amount made viable due to technological, political or economic changes since X. Resources similarly decrease due to ongoing extraction, and also change to the extent discovery and tech/pol/econ changes were not fully or correctly accounted for in original resource estimates. Around 360 billion barrels of oil have been extracted globally since 2010, and 780 billion since 1995 – these are material numbers, relative to the findings of the study.

A related weakness is that since much of the USGS data date from 1995 and 2000, they will not present an accurate picture of the present situation, even if they were adjusted for extraction and discovery since those dates. Oil extraction technologies have changed significantly in the last 20 years, including hydraulic fracturing and ultra-deepwater, and political changes have further altered the reserve base. Indeed, the study suggests at line 279 that light tight oil (i.e. oil recovered by hydraulic fracturing) is not included, which given its present importance seems a very significant omission; and I'm not really convinced by the suggestion that all oil for which data cannot be obtained should be judged unextractable (lines 284-286). Similarly, the McGlade & Ekins study is now somewhat dated, as understanding of carbon budgets has evolved significantly since then, and projected costs of renewable energy have fallen significantly (changing the balance of oil, gas and coal in future energy systems). In both respects, I shall suggest some alternative, more up-do-date data sources below.

I have two additional suggestions for the authors, which I think could significantly improve the study.

First, I am surprised the study focuses on limiting temperature increase to 2°C (with 50% probability), a somewhat outdated target that has less relevance to contemporary scientific and policy discussions. I would recommend instead focusing on the 1.5°C target of the Paris Agreement, or at least on “well below” 2°C. Well below 2°C is commonly represented in the literature either as limiting warming to below 2°C with high probability (e.g. 83%), or as limiting warming to below 1.7°C or 1.8°C with 50% probability. The Working Group I report of the IPCC's Sixth Assessment Report gives estimates for corresponding carbon budgets, and indeed I think the IPCC report – as a consensus assessment of the state of the art – would be a better source for carbon budget estimates than the single paper cited. Relatedly, the study takes McGlade & Ekins (2015) as its starting point for estimates of the total volume of burnable oil; it would be better to use Welsby et al (2021) for this purpose, which is essentially an update of McGlade & Ekins and an upgrade to 1.5°C ambition.

Second, I would recommend focusing on oil reserves. While the study generally refers to “resources”, in fact it is addressing some quantity in-between reserves and resources, namely “[resources] that had the potential to be added to reserves from 1995 to 2025” (lines 262-3), plus undiscovered oil resources as at 2012. While the way the various USGS datasets are combined isn’t entirely clear in the Methods, they would appear to exclude discovered non-US resources that USGS judged unlikely to be upgraded to reserves, for example.

In my view, reserves would be a more instructive measure, since (a) the term is well-defined and for much of the world its use and measurement are well-regulated, and (b) there is a good chance reserves will in fact be extracted absent policy intervention, whereas resource estimates are inherently more speculative (especially undiscovered resources) and are less closely tied to future prospects of extraction. A focus on reserves could also add greater resolution to the study, rather than having to assume that resources are evenly spread across each geological region (lines 471-473); indeed the authors note that limited resolution is one of the study’s weaknesses (lines 309-313).

Three options for collecting reserves data would be (i) to use a commercial source of data e.g. the Rystad UCube (<https://www.rystadenergy.com/energy-themes/oil-gas/upstream/u-cube/>) is very good, though expensive; (ii) to use a free online resource such as the World Oil Map (<https://www.oilmap.xyz/>) or the Global Registry of Fossil Fuels (<https://dev.fossilfuelregistry.org/>); or (c) to gather data bottom-up, from country and company sources.

I have some smaller general observations:

- The study builds on Codato et al (2019), which focuses only on the Amazon region. The expansion to a global scope is a valuable addition, though Codato et al (2019) focus on reserves, with data obtained bottom-up for each country from a literature review and data mining, and in that respect the present paper is weaker in my view, for the reasons above. This study includes Codato et al as a brief general reference on fossil fuel supply policy (ref. 31); I would like to see it discussed more explicitly, at least in the Methods section, since the approach is very similar.
- The introduction spends many more words than needed describing the well-known discrepancy between fossil fuel reserves/resources and carbon budgets. In contrast, the key question that sets up the whole relevance of this study – which of those reserves/resources should be allocated as burnable versus unburnable – is dealt with very briefly at lines 65-67. I would recommend expanding this, as the five cited papers take different approaches, each with strengths and weaknesses. This study appears to propose taking no account of relative costs of production (other than in the allocation between the three fossil fuels) and focusing mainly or solely on environmental and social impacts. It is far from clear how this could work in practice; an issue some of the cited papers explore. Most of these papers propose that fastest phaseout of fossil fuel production should be in the Global North, though Muttitt and Kartha

(2020) give support for excluding production where environmental justice is violated locally. Two other relevant works to consider are Armstrong (2019 <https://doi.org/10.1177/0032321719868214>) and Moss (2016 <https://doi.org/10.1080/10361146.2016.1200533>)

- It is unclear to me why urban areas (and their 10km surrounding radius) are treated as exclusion zones, while population density is only a secondary criterion for prioritising the non-excluded areas. Why not use a threshold population density to judge exclusion zones? Urban areas are defined as being built upon, but this could include an industrial park or collection of factories, which would not be such a bad place to produce. More problematic is production in densely populated rural areas, especially where people depend on local subsistence agriculture or fishing (e.g. the Niger Delta). Another possibility might be to exclude production within (say) 1km of any residence, but I guess the data on this may not be available for all countries?

- It seems that this urban exclusion zone, combined with the lack of resolution in the resource data (above), significantly shapes and potentially distorts the results, given that 36% of the unburnable oil occurs in the Zagros fold, Mesopotamian Foredeep and Rub al-Khali basins (Supplementary Table 2). These are geologically prodigious basins, and so I assume these exclusions are due to even spreading of resources across the areas of the basins, including to the cities and 10km buffer zones therein. However, most of the oilfields in these basins are not close to urban areas.

- While there is a clear case for excluding areas with presence of Indigenous peoples in voluntary isolation, it would be good to also see broader reference to Indigenous peoples, and the particular impacts of extraction they suffer. There is a good case, for example, to declare exclusion zones in territories of Indigenous peoples that have not given free, prior, informed consent for extraction according to the UN Declaration on the Rights of Indigenous Peoples.

And some more specific issues, especially related to preciseness of language:

- The term “opportunity cost” is used incorrectly throughout to mean production cost.
- The units are given as Gbl (billion barrels); bbl is the more common abbreviation for barrels, hence Gbbl (or more commonly still, bn bbl, since barrels are a non-SI unit).
- The abstract at lines 15-18 seems to adopt different framing from the main study. The abstract adopts the framing that environmentally/socially sensitive oil is not needed because enough oil is available elsewhere to meet demand in a 2C scenario. The main study rather frames it the other way around: environmentally/socially sensitive oil as the first priority for climate policies that restrict extraction.
- The goals of the Paris Agreement include “pursuing efforts” to limit warming to 1.5°C, which is stronger than limiting warming “preferably” to 1.5°C (line 28).

- At 38-39, correct to: “The unabated exploitation of all of the world’s existing fossil fuel reserves is incompatible with achieving the Paris Agreement goals”.
- At line 49, correct to: “There is currently a surge in interest in supply-side climate policies” – with the partial exception of Erickson et al, the cited works relate to proposals rather than actual policies, and where policies are occurring more concertedly (e.g. the Beyond Oil and Gas Alliance) they remain somewhat marginal.
- At line 55, reference 20 doesn’t call for a managed decline in fossil fuel supply, although a reader could perhaps infer that from the report. Such calls are somewhat nascent in the scientific and grey literature, but better citations here would be references 19, 25, 26 or 27.
- At line 57, references 12 and 20 don’t really engage in the question of selecting which resources need to stay in the ground, and reference 1 only from a cost perspective.
- At lines 62-63, it would be better to say “because of their relatively higher extraction costs and carbon content”. Better still would be “in order to limit warming to 2°C at the lowest overall cost”. (The cost optimisation is across the whole energy system). Similarly, at lines 293-294, it’s not really accurate to describe the cost-optimisation as concerning “amounts of rents generated by the extraction of the resources”.
- At lines 71-73, France and New Zealand don’t belong in this list, because their moratoria were motivated by climate change rather than local biodiversity or socio-environmental values. Costa Rica is a bit of a mixed case, but these days climate change is the primary motivation there too.
- At lines 278-279, I think it is wrong to explain the USGS estimate being smaller than McGlade & Ekins’ in terms of types of oil not included; a more important reason is that the USGS estimate is limited to resources with the potential to become reserves by 2025 plus undiscovered resources (see above), whereas McGlade & Ekins refer to all resources.
- At lines 291-293, the statement that “our identification of exclusion zones and prioritization of unburnable oil resources is based not solely on socio-environmental criteria, but also economic ones” is overstated, as the economic optimisation relates only to the balance between oil, gas and coal, not to the prioritisation or exclusion within the oil portion.
- At line 386, I would like to see literature references to support the use of a 10km radius around urban areas as “a safe zone considering health risks associated with oil extraction”.

All that said, I do believe this is a very important piece of research, and I hope the authors are not disheartened by my more critical remarks. On the contrary, I would strongly encourage them to persevere.

Reviewer #3 (Remarks to the Author):

This is a valuable contribution to the growing literature on fossil fuel supply-side approaches to climate mitigation efforts. The authors present findings that will be of significance to the academic and policy community, though they rightly note that more fine-grain analysis will be required to inform detailed policy and divestment/investment decisions. The work supports the claims provided, though it excludes from the onset unconventional oil reserves (as these are deemed unburnable in Ref. 1).

Major comments:

The restriction considerations of Indigenous peoples to those “in voluntary isolation” is understandable, but it does not reflect the widespread and often unsuccessful resistance of other Indigenous peoples to fossil fuel development on their (unrecognized) territories. It was good to see this mentioned in the conclusion as a possible application of this method.

Two reserve categories are not given specific attention, and may require a very discussion or at least mention: Oil reserves in the Arctic and Ultra-deep sea reserves, given the concerns with major spills (e.g. Deepwater Horizon).

Regarding marine biodiversity protection, there is a mention on Ln. 128 that “biodiversity hotspots are circumscribed to terrestrial areas”, which could be clarified as it may initially lead readers to think that marine biodiversity is not covered until later in the paper (through ref. 50 and the Marine Protected Areas).

More generally, the analysis does not account for the relative risks of spills between reserve types, extraction technologies, and regions. This may be something that could be suggested among the list of ‘further research’ options.

Finally, the exclusion of unconventional oil (Ln. 97-98) should perhaps be made clearer in the abstract and any communication, especially given the importance of unconventional reserves that are likely to continue being exploited (e.g. bitumen in Canada).

Minor comments:

Ln. 75-78, for impacts on fisheries and coastal communities, see for example: Andrews, N., Bennett, N. J., Le Billon, P., Green, S. J., Cisneros-Montemayor, A. M., Amongin, S., ... & Sumaila, U. R. (2021). Oil,

fisheries and coastal communities: A review of impacts on the environment, livelihoods, space and governance. *Energy Research & Social Science*, 75, 102009.

Ln. 90, it is unclear if conventional oil includes unconventional oil (as mentioned Ln. 61) as unconventional oil is not mentioned in the subsequent sentence (only unburnable coal and natural gas). But this is almost immediately clarified at the beginning of the following paragraph, so probably no need to revise statement on Ln. 90.

Ln. 395, minor typo ('datalayer').

Ln. 396 (and throughout the paper), the term Indigenous is capitalized by most Indigenous studies scholars, and people is generally used in the plural form (as done Ln. 397).

Ln. 402, minor typo (':').

Response to the referees' comments

Ms. Ref. No.: NCOMMS-21-41044-T

Title: The atlas of unburnable oil: Spatial criteria for supply-side climate policies

We would like to thank the referees for the careful reading of our manuscript and for their extensive and constructive feedback. Below we summarise their comments and present our responses. To show how the paper has changed in relation to the comments, we copy-pasted the relevant text into this document. Minor changes (which include new references) have been included only in the manuscript.

Reviewers' comments:

Reviewer #1:

#1.1

This is a really interesting paper that considers how to prioritise what conventional oil resources need to stay in the ground based on a range of social and environmental criteria, thereby creating an atlas of exclusion areas for extraction. This is a welcome addition to the literature that is often dominated by economic efficiency objectives in terms of unburnable reserves and resources.

Thank you.

#1.2

Climate ambition

The paper is framed in terms of 2°C at 50%. I understand you have chosen this as it fits with the McGlade and Ekins (M&E) paper, from which determine % unburnable resources. However, much of the current discourse is around the transition away from fossil fuels under 1.5°C. Furthermore, the problem with 2°C at 50% is that it is not consistent with the Paris Agreement (of 'well below 2°C').

The Welsby et al. 2021 paper could be considered as an alternative source, although perhaps you did not pick this up due to the timing of its publication? In that paper, unextractable estimates for conventional oil resources sit at 1823 Gb, or 71%. This is quite a lot higher than in M&E. These numbers, which are used in SI Fig 1, could be requested from the authors.

My suggestion is that this paper would have more impact if framed using stronger ambition and would be more relevant to the current debate around fossil fuel phase out.

We have re-calculated the figures according to the 1.5°C target in line with the Paris agreement. We requested the unpublished data from Welsby et al. 2021, who kindly shared them, and adjusted our analysis. Welsby et al. 2021 estimated that 1,823 Gbbls (71%) of the conventional oil resources are unextractable, and therefore, 752Gbbls (29%) could be burned. In our analysis, we have now allocated these 752Gbbls according to the proposed social and environmental criteria.

#1.3

One additional point that needs to be recognised in relation to budgets discussed in the paper; the budget quoted in M&E was for 2011-2050, while other budgets such as those from Matthews et al are from 2020-2100. You need to make that clear. Also, M&E didn't run their model with a carbon budget but used a climate module; they mention a budget in their abstract but do not report a budget derived from their own analysis. This also needs to be made clearer in your paper.

We no longer use the data from M&E 2015 and we have clarified that Welsby et al. 2021 use a global energy systems model and a 2018–2100 carbon budget of 580 GtCO₂.

The text now reads:

“Ref.(Welsby et al., 2021) found that 71% and 81% of the conventional oil and gas resources, respectively, should remain unextracted by 2050²² (Supplementary Table 1). 99% of unconventional oil resources, 93% of unconventional gas resources and 97% of coal should remain unburned by 2050 in order to limit global warming at the lowest overall cost²². These estimates were calculated using a global energy systems model and a 2018–2100 carbon budget of 580 GtCO₂²².”

#1.4

Unburnable conventional oil resources

I am little confused with the unburnable numbers that you are using. You start with total conventional oil resources of 1818 Gb but state that '611 Gbl (263 GtCO₂) out of 1,818 Gbl of conventional oil resources need to be left untapped'. How do you determine 611 Gb? This seems to assume a 33% unburnable level, which is the reserves percentage from M&E. However, I think you want to use 54%, for conventional oil resources, as per M&E, which would give you an unburnable value of 981 Gb. If I have misunderstood this, then please make your calculations and assumptions clearer, being more explicit about the numerator/denominator for these estimates.

If this does indeed need correcting, it means that you may need to re-think / redo your three additional scenarios, as the gap between that which is excluded (457 Gb) and total unburnable resources will be significantly larger.

The estimate of unburnable resources is now explained in detail in the *Methods* section that is updated in line with the data from Welsby et al. 2021:

“There are significant differences between the georeferenced resources we use, and the non-georeferenced tabular data provided by other sources and datasets. We put together georeferenced data for 1,627 Gbbl of conventional oil resources, while ref.(Welsby et al., 2021) provided an estimate of 2,576 Gbbl of conventional oil resources. These differences are explained by the dearth of accessible and up-to-date spatial data. Ref.(Welsby et al., 2021) estimated that 1,823 Gbbl (71%) of the conventional oil resources should be considered unburnable and that, therefore, 752 Gbbl (29%) could be burned,. Out of the 1,627 Gbbl of georeferenced conventional oil resources, 875 Gbbl (1,627 Gbbl minus 752 Gbbl) should be left untapped and we use socio-environmental variables to rank and prioritize these resources.”

We have also amended the main text to clarify:

“Considering the carbon budget associated with the target of 1.5°C of global warming and applying a cost-optimal distribution of the carbon budget among coal, gas, and oil (following ref. Welsby et al. 2021), 71% (752 Gbbl) of the 1,823 Gbbl conventional oil resources can be extracted. We adopt this estimate of extractable oil and out of 1,627 Gbbl of conventional oil resources we consider that 875 Gbbl need to be left untapped (see the *Methods* section for a detailed explanation).”

#1.5

Finally, given that you focus on resources, not reserves, it may be worth re-thinking the introduction that focuses on reserves and could cause confusion (lines 40-45). The focus of the paper is on conventional oil resources so perhaps worth focusing on this, in respect of this context setting section – and keeping any discussion of reserves in the Methods.

The introduction of the manuscript now focusses on resources, and we have deleted the references to reserves from the main text.

#1.6

3. Regional distribution

I think this paper would be significantly strengthened by further elaborating on the distribution of oil resources by region and making some comment on how this differs from M&E or Welsby et al. Firstly, it would be good if you were able to state the change in regional distribution of unburnable resources. Secondly, a comparison with the cost-optimal distributions from other papers would highlight which regions would be most impacted by alternative socio-environmental criteria – and reveal the tension discussed by Lenferna in his paper between efficiency and other criteria for prioritising unburnable oil.

We have included in the main text a Table with an overview of regional distribution of unburnable resources:

“Table 2 presents the regional distribution of unburnable conventional oil resources according to the exclusion criteria. While the Middle East is the region with most unburnable conventional oil in absolute

terms, the region of Developing Asia (outside of China and India) stands out with approximately 88% of the resources unburnable.”

Region	Unburnable conventional oil - by region (Gbbbl)	Regional unburnable conventional oil (%)
Africa	76.49	30.56
Australia and other OECD Pacific	4.69	51.63
Canada	0.48	2.23
China and India	17.66	32.22
Russia and former Soviet states	34.20	13.27
Central and South America	84.56	34.23
Europe	7.70	11.65
Middle East	164.16	26.49
Other Developing Asia	30.92	88.37
USA	26.26	40.00
Global	447.11	27.48

Furthermore, the Supplementary materials now include:

“When compared to the regional distribution of unburnable conventional resources proposed by REF Welsby et al. 2021, the exclusion zones produce lower proportions of unburnable conventional oil resources (as expected since globally they also concern less resources) with the greatest difference for Canada (2% and 82% unburnable conventional resources in our analysis and Welsby et al. 2021, respectively) and the exception of Developing Asia (outside of China and India; 88% and 48% unburnable conventional resources in our analysis and Welsby et al. 2021, respectively). The regions with large differences are indicative of a divergence between the socio-environmental criteria we employed to identify unburnable resources and the economic ones used by Welsby et al. 2021.”

Other briefer comments:

- Line 9-12. The way this is written sounds like M&E assumed 1370 MtCO2 budget, which they didn’t

We now refer to Welsby et al. 2021 and to their carbon budget, see #1.3.

- Line 17. The figure of 611 needs checking – and amending if necessary

See response to comment #1.4

- Line 31-35: for latest budget, better to refer to recent AR6 WG1

We now refer to the carbon budget from the AR6 and we have included a reference to (IPCC, 2022).

- Line 40-41: reference 6 says nothing about current reserves

We have eliminated reference 6.

- Lines 40-45: the discussion on reserves here could confuse the reader. The focus of the paper is on conventional resources so perhaps worth focusing on this, in respect of this context setting section.

See the response to comment #1.5

- Line 49: the latest Production Gap report provides a useful listing of the various supply side efforts around the world.

We now included a reference to the Production Gap Report (SEI, IISD, ODI, E3G, and UNEP, 2021) when referring to the country initiatives to limit the supply of fossil fuels.

- Lines 51-55: it would be worth mentioning the recent BOGA initiative that gained much attention at COP26

We now also mention the Beyond Oil and Gas Alliance (BOGA, 2022).

- Line 88-91: will you publish the datasets, including spatial layers? This would be important for allowing others to develop and use such information

We will follow the FAIR principles (i.e. Findable, Accessible, Interoperable and Reusable data) and the principles of EU Open Science Policy, and we will publish datasets in the Digital Repository of our University (further details withheld to ensure anonymity of this submission).

- Line 216-217: implications also for donor and investment community, providing financing

The conclusions now include:

“It also makes it possible for energy corporations, governments and, more generally, investors to minimize the risks of stranded assets”

- Line 280/296: M&E didn't use a carbon budget but rather a climate module to determine unburnable fossil fuels under a 2C target

See our response to comment #1.2.

- Line 282-283: You state ‘This means that 611 Gbbl (263 GtCO₂) of our 1,818Gbbl (781 GtCO₂) of georeferenced conventional oil resources should be left untapped.’ You need to provide information on how you get to 611 Gb (as per earlier comments)

See response to comment #1.4

- Line 291-295: how does the cost-optimal distribution inform your analysis? Can you elaborate on this? Also, do you start with the regional estimates of unburnable conventional oil resources from M&E for you analysis or simply take the global total, developing the distribution from that basis?

Our analysis now takes the global total distribution of unburnable fuels among coal, gas and oil from Welsby et al 2021 . We have now clarified this in the *Methods* section:

“To identify the unburnable oil resources whose conservation would generate substantial collateral socio-environmental benefits, our spatial analysis builds upon the global cost-optimal distribution of unburnable fuels among coal, gas and oil proposed by ref. Welsby et al. 2021 (i.e. a key assumption of our spatial analysis is that, as set by Welsby et al. 2021, 71% of the conventional oil resources should remain unextracted by 2050, as well as 81% of conventional gas resources, 99% of unconventional oil resources, 93% of unconventional gas resources and 97% of coal).”

Reviewer #2:

#2.1

This study makes a valuable contribution to debates about “unburnable carbon” and about co-benefits of decarbonisation.

It is well known that the world’s fossil fuel reserves significantly exceed what can be burned while achieving the Paris climate goals, and therefore only a portion of reserves can be safely extracted. The key question is: which portion? This study answers that it should be the portion where environmental and social costs are lowest, and proposes a method for assessing oil provinces accordingly. By marrying spatial data on oil resources, on areas of biodiversity importance and on potential social impacts, the study shows how far environmental and social criteria can be used to shape decisions on which fossil fuels should remain unextracted.

Thank you.

#2.2

However, I believe the study requires some further work before publication.

In particular, I think there is a methodological flaw, namely neglecting or mischaracterising the time dimension of oil production. McGlade & Ekins estimate an amount of oil that can be consumed between 2010 and 2050 in a 2°C scenario; the authors appear to interpret this wrongly as an amount that can be consumed from the present day until the ultimate end of oil production (e.g. the discussion of carbon budgets is effective from 2020). The geographical location of the oil resources is taken from USGS resource estimates from 1995, 2000 and 2012; these too are apparently taken to represent the present picture. In reality, reserves and resources change over time: today’s reserves are equal to the reserves at prior date X, minus the amount extracted since X, plus the amount discovered since X, plus the amount made viable due to technological, political or economic changes since X. Resources similarly decrease due to ongoing extraction, and also change to the extent discovery and tech/pol/econ changes were not fully or correctly accounted for in original resource estimates. Around 360 billion barrels of oil have been extracted globally since 2010, and 780 billion since 1995 – these are material numbers, relative to the findings of the study.

Thank you for raising these points that need to be clarified. We have amended the text to make more explicit the time dimension of oil production and the estimates of unextractable oil.

On the one hand, we have clarified the time frame of the model used by Welsby et al 2021 to estimate the unextractable resources:

“Ref.(Welsby et al., 2021) found that 71% and 81% of the conventional oil and gas resources, respectively, should remain unextracted by 2050²²”.

“It is important to note that continued use of fossil fuels after 2050 (strongly dependent on Carbon Dioxide Removals (CDR) and mainly for feedstocks in the petrochemical sector, and as fuel in the aviation sector in the case of oil) sees these estimates diminish by 2100 (for instance, estimates of unextractable conventional oil and gas resources are reduced to 54% and 76% respectively) (Welsby et al., 2021).”

On the other hand, we have updated the resource estimates used (see response to comment #2.3) and included the following text acknowledging the limitations of the available data on oil resources:

“However, it is important to note that these estimates of the global conventional oil resources are subject to some degree of uncertainty (as demonstrated by the variation of the estimates over time and across different sources) and are subject to obsolescence due to ongoing extraction –approximately 33 billion barrels of oil have been extracted per year globally since 2010 (BP 2022).”

#2.3

A related weakness is that since much of the USGS data date from 1995 and 2000, they will not present an accurate picture of the present situation, even if they were adjusted for extraction and discovery since those dates. Oil extraction technologies have changed significantly in the last 20 years, including hydraulic fracturing and ultra-deepwater, and political changes have further altered the reserve base. Indeed, the study suggests at line 279 that light tight oil (i.e. oil recovered by hydraulic fracturing) is not included, which given its present importance seems a very significant omission; and I’m not really convinced by the suggestion that all oil for which data cannot be obtained should be judged unextractable (lines 284-286). Similarly, the MacGlade & Ekins study is now somewhat dated, as understanding of carbon budgets has evolved significantly since then, and projected costs of renewable energy have fallen significantly (changing the balance of oil, gas and coal in future energy systems). In both respects, I shall suggest some alternative, more up-do-date data sources below.

We thank the reviewer for pointing these important issues. In the new version we have either updated and improved the datasets on resources or more explicitly recognized the limitations of the study:

1.- We now use data from Welsby et al 2021 rather than MacGlade and Ekins 2015, using the new balance of oil, gas and coal in future energy systems that incorporate the changes in energy costs. See the response to comment #1.3

2.- We have revised our georeferenced global database on conventional oil resources, improving on the limitations highlighted by the reviewer. We have now reviewed all the World and Gas Resource Assessments (n= 142) for undiscovered conventional oil published by the USGS from 2012 to September 2022. Out of the 142 reviewed reports, 33 are re-assessing undiscovered oil and gas resources for 93 sedimentary basins (6 US basins and 87 outside US). The new data have been used to update resources volumes in our georeferenced global database and we have redone our spatial analysis. Although the limitation on the remaining discovered resources remains (we still use data from the WPA 2000, as USGS has not updated or assessed reserves since the WPA 2000), the inclusion of the new assessments on undiscovered resources allows us to present a more accurate picture of the present situation, taking into account new oil extraction technologies. We have clarified the remaining limitations and changed the text when needed in the *Methods*. Now it reads:

“For that purpose, we compiled a dataset of global conventional oil resources based on data from the United States Geological Survey (USGS). The dataset was put together following the method described by ref. (Butt et al., 2013). For world conventional oil resources, we used data from the World Oil and Gas Assessments produced by the USGS World Energy Project, in particular the USGS World Petroleum Assessment (WPA) 2000(USGS, 2000) and 2012(USGS, 2012a) and, 27 regional USGS assessment reports on undiscovered oil and gas resources in priority geologic provinces in the World published between 2012 and September 2022 (<https://www.usgs.gov/centers/central-energy-resources-science-center/science/world-oil-and-gas-resource-assessments#publications>). The USGS WPA 2000 provides estimates of the quantities of conventional oil resources outside the United States that had the potential to be added to reserves from 1995 to 2025. We used spatial and tabular data on remaining oil and remaining NGL (i.e. quantities of conventional oil and NGL excluding reported cumulative volume of oil and NGL that had been already produced) from the WPA 2000 (“wep_prvg.shp” geospatial data set). From the USGS WPA 2012, we have retrieved estimates of the quantities of undiscovered conventional oil resources. Specifically, we used spatial and tabular data on undiscovered technically recoverable oil resources (i.e., the oil that could be produced using available technology and industry practices regardless of economic or accessibility considerations) from the WPA 2012 (“Province Summary.xls”). When individual regional reports for specific sedimentary basins have been issued after 2012, undiscovered conventional oil volumes from the WPA 2012 have been updated. Thus, conventional oil resources volumes have been calculated by adding remaining discovered conventional oil resources as in the WPA 2000, undiscovered conventional oil resources as in the WPA 2012, and individual USGS assessments for sedimentary basins from 2012 onwards.

For USA conventional oil resources, we used data from National Oil and Gas Assessments (NOGA) produced by the USGS. We used tabular data on oil conventional oil resources volumes for USA geological provinces from the USGS NOGA Resources Update (March, 2013) (USGS, 2013) and 6 USGS national assessment reports on undiscovered oil and gas resources published between 2012 and September 2022. The spatial data for USA geological provinces were acquired from the USGS NOGA 1995 (USGS, 1995) and the NOGA Province Boundaries update from 2012 (USGS, 2012b).

These are the most accurate and up to date open-access and available georeferenced datasets on conventional oil resources at the global level. However, it is important to note that these estimates of the global conventional oil resources are subject to some degree of uncertainty (as demonstrated by the variation of the estimates over time and across different sources) and are subject to obsolescence due to ongoing extraction –approximately 33 billion barrels of oil have been extracted per year globally since 2010 (BP 2022).”

We also agree with the reviewer that the statement “Any other conventional oil resource without georeferenced data not considered in this study (i.e. oil reserve growth, NGL reserve growth, and light tight oil) could not be burned in order to limit global warming to well below 2°C” (line 283- 286), could lead to misinterpretation. We have now rephased it:

“If any of the other conventional oil resources not considered in this study (because of the unavailability of georeferenced data) are burned, a larger portion of the conventional oil resources for which we have georeferenced data should be left untapped in order to limit global warming to 1.5°C.”

#2.4

I have two additional suggestions for the authors, which I think could significantly improve the study.

First, I am surprised the study focuses on limiting temperature increase to 2°C (with 50% probability), a somewhat outdated target that has less relevance to contemporary scientific and policy discussions. I would recommend instead focusing on the 1.5°C target of the Paris Agreement, or at least on “well below” 2°C. Well below 2°C is commonly represented in the literature either as limiting warming to below 2°C with high probability (e.g. 83%), or as limiting warming to below 1.7°C or 1.8°C with 50% probability. The Working Group I report of the IPCC’s Sixth Assessment Report gives estimates for corresponding carbon budgets, and indeed I think the IPCC report – as a consensus assessment of the state of the art – would be a better source for carbon budget estimates than the single paper cited. Relatedly, the study takes McGlade & Ekins (2015) as its starting point for estimates of the total volume of burnable oil; it would be better to use Welsby et al (2021) for this purpose, which is essentially an update of McGlade & Ekins and an upgrade to 1.5°C ambition.

This resonates with the comment from referee 1 (#1.2- and 1.3) and the manuscript now focuses on the 1.5°C target based on Welsby et al 2021. See the response to comments #1.2 and 1.3 above.

#2.5

Second, I would recommend focusing on oil reserves. While the study generally refers to “resources”, in fact it is addressing some quantity in-between reserves and resources, namely “[resources] that had the potential to be added to reserves from 1995 to 2025” (lines 262-3), plus undiscovered oil resources as at 2012. While the way the various USGS datasets are combined isn’t entirely clear in the Methods, they would appear to exclude discovered non-US resources that USGS judged unlikely to be upgraded to reserves, for example.

In my view, reserves would be a more instructive measure, since (a) the term is well-defined and for much of the world its use and measurement are well-regulated, and (b) there is a good chance reserves will in fact be extracted absent policy intervention, whereas resource estimates are inherently more speculative (especially undiscovered resources) and are less closely tied to future prospects of extraction. A focus on reserves could also add greater resolution to the study, rather than having to assume that resources are evenly spread across each geological region (lines 471-473); indeed the authors note that limited resolution is one of the study’s weaknesses (lines 309-313).

Three options for collecting reserves data would be (i) to use a commercial source of data e.g. the Rystad UCube (<https://www.rystadenergy.com/energy-themes/oil-gas/upstream/u-cube/>) is very good, though expensive; (ii) to use a free online resource such as the World Oil Map (<https://www.oilmap.xyz/>) or the Global Registry of Fossil Fuels (<https://dev.fossilfuelregistry.org/>); or (c) to gather data bottom-up, from country and company sources.

We agree with the reviewer that reserves are more likely to be extracted than resources and it would be interesting to see an Atlas of unburnable oil reserves. However, we also consider that an analysis of resources at the level of sedimentary basins is equally of interest, especially within the current increase of interest on supply-side climate policies, to guide, for instance, investment in oil exploration. We

acknowledge that our resource estimates are imperfect (see comment #2.3 above). At the same time, reserve estimates fluctuate over time depending on prices, the cost of available technologies, the development of new oil extraction technologies, and new discoveries, and strategic overestimation by rights holders (Green 2022; Laherrère et al. 2022). In turn, there is much uncertainty on future energy prices which heavily depend on some key political choices, including the climate mitigation policies adopted by governments around the world. Ultimately, reserves depend crucially on climate policies.

From the policy perspective, halting exploration for new fossil fuel resources to be added to the reserve base is a first logical step towards the fulfilment of the Paris Agreement commitments. We consider that that an analysis focusing on reserves is an important next step in the research on unextractable fossil fuels. As we mention in the manuscript, “the use of more fine-grained data on reserves and specific reservoirs and oil fields could produce atlases of unburnable oil that would be appropriate for local and regional policymaking”. However, there are some important difficulties for such an analysis, as “most of these data are inaccessible to the scientific community.” We tried to get access to another private global spatial database on oil and gas reserves (Wood MacKenzie’s Lens, former Petroview), but they only provide services to energy corporations making them effectively unavailable to research institutions. Rystad Ucube database is proprietary and prohibitively expensive. Open-access databases such as the World Oil Map or the Global Registry of Fossil Fuels have important limitations on the data quality. The World Oil Map does not provide data on the volumes of reserves of oil blocks or deposits. It “just” provides global data on location of deposits and oil blocks, operating companies and years of operation. Similarly, the Global Registry of Fossil Fuels does provide data aggregated by country, not per fields or deposits. The last alternative, i.e. to gather global data bottom-up, from country and company sources, is an insurmountable task for our research group and motivates future collaborative projects such as the mentioned above.

Finally, one of the main aims of our manuscript is to “propose a methodology to identify and prioritize unburnable fossil fuels [] according to environmental and social criteria”. Our methodology could later be used to prepare atlases for unburnable conventional oil reserves and also for unburnable coal and natural gas.

I have some smaller general observations:

#2.6

- The study builds on Codato et al (2019), which focuses only on the Amazon region. The expansion to a global scope is a valuable addition, though Codato et al (2019) focus on reserves, with data obtained bottom-up for each country from a literature review and data mining, and in that respect the present paper is weaker in my view, for the reasons above. This study includes Codato et al as a brief general reference on fossil fuel supply policy (ref. 31); I would like to see it discussed more explicitly, at least in the Methods section, since the approach is very similar.

We have reviewed existing studies to ensure that we acknowledge previous research that used similar approaches. To the best of our knowledge, RAISG (*Red Amazónica de Información Socioambiental Georeferenciada*) produced the first study using socio-environmental criteria to determine unburnable fossil fuels (RAISG, 2012. Amazonía bajo presión. Sao Paolo:

<https://www.raisg.org/es/publicacion/amazonia-bajo-presion/>). RAISG 2012 studied the overlap between oil and gas infrastructure (pipelines, wells, seismic lines, oil and gas blocks) and indigenous territories and protected areas in the Amazon. Codato et al 2019 use a similar approach. However, RAISG 2012 and Codato et al 2019 do not analyze the estimates of oil and gas reserves (or resources), nor do they produce a prioritization analysis. While ours is not simply an expansion to a global scope of RAISG 2012 and Codato et al 2019, these are very relevant contributions since they highlighted that constraining fossil fuel supply can generate additional sustainability benefits because of the local socio-environmental impacts of extraction. Harfoot *et al.* 2018 (Present and future biodiversity risks from fossil fuel exploitation. *Conserv. Lett.* **11**, e12448) is also a relevant study and we now cite all of them in the main text.

#2.7

- The introduction spends many more words than needed describing the well-known discrepancy between fossil fuel reserves/resources and carbon budgets. In contrast, the key question that sets up the whole relevance of this study – which of those reserves/resources should be allocated as burnable versus unburnable – is dealt with very briefly at lines 65-67. I would recommend expanding this, as the five cited papers take different approaches, each with strengths and weaknesses. This study appears to propose taking no account of relative costs of production (other than in the allocation between the three fossil fuels) and focusing mainly or solely on environmental and social impacts. It is far from clear how this could work in practice; an issue some of the cited papers explore. Most of these papers propose that fastest phaseout of fossil fuel production should be in the Global North, though Muttitt and Kartha (2020) give support for excluding production where environmental justice is violated locally. Two other relevant works to consider are Armstrong (2019 <https://doi.org/10.1177/0032321719868214>) and Moss (2016 <https://doi.org/10.1080/10361146.2016.1200533>)

Thanks for this observation. Indeed, this study uses the costs of production to determine the percentage of oil resources (when compared to other fossil fuel resource categories) to be left unextracted taking the allocation from Welsby et al. 2021 as the starting point. The objective is to show that supply side climate policies have a potential to produce substantial collateral benefits if associated with policies that would prioritize the phase out of fossil fuels extraction/exploration in areas where resources coincide with relevant socio-environmental values. There is already a more focused literature on how this could look in practice and how the global community could move towards a regime of supply side climate policies. This said, the point that we could have elaborated further is well taken and we now discuss some of the policies/principles that could underpin such a regime.

Now the text includes:

“The allocation of the remaining fossil fuels that can be extracted is a morally and politically contentious issue (Armstrong, 2020) that is entwined with the issue of compensation (Orta-Martínez et al., 2022). In particular, developed countries have historically contributed the most to accumulated greenhouse gas emissions generating an ecological debt (Martinez-Alier, 2021) and fossil fuel rents could contribute towards developing countries’ right to develop (Armstrong, 2020). While several ethical considerations suggest that developed countries are those who should leave their fossil fuel reserves underground, considerations of economic efficiency would instead suggest that those resources whose extraction

generates lower rents (because of relatively higher extraction, transportation and transformation costs, or lower economic value of the resources) should be left untapped, and as we show here the different socio-environmental impacts of extraction in different locations can also produce alternative distributions of unburnable fossil fuels (Muttitt & Kartha, 2020; Lenferna, 2018; Pye et al., 2020). Also, (partial) compensation could be considered a condition for the political feasibility of any multilateral international agreement to keep fossil fuels underground (Pellegrini and Arsel, 2022) and the pathway to an agreement could include intermediary steps to strengthen international norms against fossil fuels (Green, 2018) and club arrangements to pave the way to the agreement (van Asselt & Newell, 2022)."

#2.8

- It is unclear to me why urban areas (and their 10km surrounding radius) are treated as exclusion zones, while population density is only a secondary criterion for prioritising the non-excluded areas. Why not use a threshold population density to judge exclusion zones? Urban areas are defined as being built upon, but this could include an industrial park or collection of factories, which would not be such a bad place to produce. More problematic is production in densely populated rural areas, especially where people depend on local subsistence agriculture or fishing (e.g. the Niger Delta). Another possibility might be to exclude production within (say) 1km of any residence, but I guess the data on this may not be available for all countries?

Thanks for this comment. We use the variable 'urban areas' as a proxy of population density. We agree that ideally, we would be excluding the most populated areas (both rural and urban) of the world from oil extraction. To our knowledge, the MODIS 500-m global map of urban extent produced by the University of Wisconsin-Madison, Boston University and the MODIS Land Group is the best global dataset available for this purpose. This dataset, which exploits remotely sensed imagery as input data, defines 'urban areas' as places larger than 1km² that are dominated (coverage greater than or equal to 50%) by the built environment (i.e., non-vegetative, human-constructed elements, such as buildings, roads, runways, etc.). Although these 'urban areas', as mentioned by the reviewer, include industrial areas, the database is, to our knowledge, the best proxy for global data on residence, at such fine resolution, as they include small towns and rural villages. Residential data at a similar or higher resolution is not available at the global level. Furthermore, the widely used FAO rural population density map is based on estimates of global population distribution, extrapolating data from national censuses.

We have articulated this explanation in the 'Methods' section:

"Regarding data on urban areas, we used the MODIS 500-m global map of urban extent produced by the University of Wisconsin-Madison, Boston University and the MODIS Land Group (Schneider et al., 2009, 2010). 'Urban areas' are identified through remote sensing based on physical attributes (not on population densities). The 'MODIS 500-m global map of urban extent' defines 'urban areas' as pixels that are dominated by the built environment. 'Built environment' includes all non-vegetative, human-constructed land covers, such as buildings, roads, runways, etc., and 'dominated' implies coverage greater than or equal to 50% of the pixel. All these areas, including a 10 km buffer around them, were designated as irreconcilable with oil extraction. We assumed these areas to be densely populated areas (both rural and urban). The 10km buffer was set considering health risks associated with oil extraction.

For the areas beyond the 'built environment', we used a dataset for rural population density, the Rural population density map of the Food Insecurity, Poverty and Environment Global GIS Database (FGGD) (FAO, 2007). FGGD is a global database maintained by the Food and Agricultural Organization (FAO) to analyse food insecurity and poverty in relation to the environment. The FGGD rural population density map provides estimates of the global population distribution in 2015. It is a global raster datalayer with a resolution of 5 arc-minutes. Each pixel classified as non-built environment by the urban area boundaries map contains the number of persons per square kilometer, aggregated from the 30 arc-second datalayer. The method used by FAO to generate this datalayer is described in ref.(Mirella Salvatore et al., 2005)."

#2.9

- It seems that this urban exclusion zone, combined with the lack of resolution in the resource data (above), significantly shapes and potentially distorts the results, given that 36% of the unburnable oil occurs in the Zagros fold, Mesopotamian Foredeep and Rub al-Khali basins (Supplementary Table 2). These are geologically prodigious basins, and so I assume these exclusions are due to even spreading of resources across the areas of the basins, including to the cities and 10km buffer zones therein. However, most of the oilfields in these basins are not close to urban areas.

We appreciate the reviewer's comment. The lack of access to spatial data on oil reserves and resources at a higher resolution is the main data limitation of the present work, as we acknowledge:

"Regarding our spatial unit of analysis, we have used the 246 Assessment Units of the WPA 2000 (USGS, 2000), located in one of the 128 geologic provinces of the world, and 72 USA geological provinces (USGS, 1995, 2012b). Geological provinces are USGS-defined areas having characteristic dimensions of hundreds to thousands of kilometers encompassing a natural geologic entity (i.e. sedimentary basin, thrust belt, delta), or some combination of contiguous geologic entities. We are aware that the use of finer spatial units of analysis, such as oil fields or reservoirs, would be a significant methodological improvement, because it would bring mapping to scales comparable with regional decisions on which oil fields and reservoirs should be kept untapped. However, these data are inaccessible for scientific purposes, or are only provided for a substantial price."

Indeed, the top-priority unburnable conventional oil resources located in small portions of sedimentary basins from the Arabian Peninsula (Mesopotamian Foredeep Basin, 66Gbbbl; Rub Al Khali Basin, 23Gbbbl) and the Iranian Zagros mountains (Zagros Fold Belt, 77Gbbbl), might be influenced by the use of geological provinces. Now the text includes:

"Conventional oil resources present at each geologic assessment unit were divided and proportionally assigned to each pixel of the geologic assessment unit according to its surface area. This is, in fact, the main limitation of the data used here."

This said, here, we provide to the reviewer a closer view of the exclusion zones in the Zagros fold, Mesopotamian Foredeep and Rub al-Khali basins. Globally, these exclusion zones are due to the overlap of the sedimentary basins with biodiversity hotspots and Natural Protected Areas, and to a minor degree with the urban areas and the 10km buffer zones:

Zagros Fold Belt:

Total overlap: 60,2% (50,5% with biodiversity hotspots, 14,75% with urban areas, including the 10km buffer).

Rub Al Khali Basin:

Total overlap: 16,6% (12,2% with Natural Protected Areas –WDPA-, 3,6% with marine richness centers of endemic species, and 0,9% with urban areas, including the 10km buffer).

Mesopotamian Foredeep basin:

Total overlap: 21,8% (15,8% with urban areas, including the 10km buffer, 6,3% with Natural Protected Areas –WDPA-, and 0,3% with marine richness centers of endemic species.

#2.10

- While there is a clear case for excluding areas with presence of Indigenous peoples in voluntary isolation, it would be good to also see broader reference to Indigenous peoples, and the particular impacts of extraction they suffer. There is a good case, for example, to declare exclusion zones in territories of Indigenous peoples that have not given free, prior, informed consent for extraction according to the UN Declaration on the Rights of Indigenous Peoples.

We do agree with the reviewer on the fact that there is a good case to include the territories of indigenous groups that reject oil exploration and exploitation as ‘exclusion areas’. While there is no available global data on this, we now discuss further the impacts of oil extraction in indigenous territories and specifically the issue of free, prior, informed consent. We have now included a paragraph on this in the manuscript:

“Oil extraction in indigenous territories has been often associated with direct socio-environmental consequences and health risks (Orta-Martínez et al 2007, Orta-Martínez et al 2018, O’Callaghan et al 2021). Indigenous people have quite often opposed oil extraction in their territories (Orta-Martínez et al. 2010) and their right to Free, Prior and Informed Consent, established by the UN Declaration on the Rights of Indigenous Peoples, has not been secured in most of the oil projects that overlap with indigenous territories (Greenspan et al 2015) and the absence of consent could be used to demarcate unextractable oil resources.”

And some more specific issues, especially related to preciseness of language:

- The term “opportunity cost” is used incorrectly throughout to mean production cost.

Thanks for flagging this, the two references to “opportunity cost” have now been corrected to “production cost”.

- The units are given as Gbl (billion barrels); bbl is the more common abbreviation for barrels, hence Gbbl (or more commonly still, bn bbl, since barrels are a non-SI unit).

We prefer to use ‘gigabarrel’ instead of ‘billion barrels’ to avoid any possible confusion between long and short scale billions. Regarding the use of Gbl or Gbbl, we adopted the suggestion of the referee and use Gbbl.

- The abstract at lines 15-18 seems to adopt different framing from the main study. The abstract adopts the framing that environmentally/socially sensitive oil is not needed because enough oil is available elsewhere to meet demand in a 2C scenario. The main study rather frames it the other way around: environmentally/socially sensitive oil as the first priority for climate policies that restrict extraction.

Thanks for highlighting this point that might disorient the reader. It is our objective to highlight both: a) that the conservation of oil resources coinciding with high priority exclusion zones can be excluded from extraction since they are well-within the required amount of unburnable oil resources to halt global warming, and also b) that the unburnable resources can be prioritized to reap collateral socio-environmental benefits. We also note that these points became only stronger because more oil needs to be kept under the soil in line with the 1.5C scenario if compared to the 2C scenario. Now the main text also reflects this perspective and we included:

“These zones overlap with 447 Gbl of oil resources that by themselves are insufficient to meet climate policy targets, implying that they can be kept entirely off limits to contribute to climate policy and still other oil resources will also need to be left under the soil.”

- The goals of the Paris Agreement include “pursuing efforts” to limit warming to 1.5°C, which is stronger than limiting warming “preferably” to 1.5°C (line 28).

We now write “The Paris Agreement aims to limit global warming to well below 2°C, pursuing efforts to limit it to 1.5°C, compared to pre-industrial levels.”

- At 38-39, correct to: “The unabated exploitation of all of the world’s existing fossil fuel reserves is incompatible with achieving the Paris Agreement goals”.

Addressed as suggested.

- At line 49, correct to: “There is currently a surge in interest in supply-side climate policies” – with the partial exception of Erickson et al, the cited works relate to proposals rather than actual policies, and

where policies are occurring more concertedly (e.g. the Beyond Oil and Gas Alliance) they remain somewhat marginal.

Addressed. Now it reads:

“There is currently a surge in interest by academics and policy-makers on supply-side climate policies to limit fossil fuel production”.

- At line 55, reference 20 doesn't call for a managed decline in fossil fuel supply, although a reader could perhaps infer that from the report. Such calls are somewhat nascent in the scientific and grey literature, but better citations here would be references 19, 25, 26 or 27.

Addressed as suggested.

- At line 57, references 12 and 20 don't really engage in the question of selecting which resources need to stay in the ground, and reference 1 only from a cost perspective.

Addressed as suggested.

- At lines 62-63, it would be better to say “because of their relatively higher extraction costs and carbon content”. Better still would be “in order to limit warming to 2°C at the lowest overall cost”. (The cost optimisation is across the whole energy system). Similarly, at lines 293-294, it's not really accurate to describe the cost-optimisation as concerning “amounts of rents generated by the extraction of the resources”.

Addressed as suggested also by referee 1, see #1.3.

- At lines 71-73, France and New Zealand don't belong in this list, because their moratoria were motivated by climate change rather than local biodiversity or socio-environmental values. Costa Rica is a bit of a mixed case, but these days climate change is the primary motivation there too.

Thanks, indeed, the France and New Zealand cases are motivated by and large by climate change concerns, and we deleted them from the list. With Costa Rica, we would argue that the motivation of the decision is mixed and very much related to both the environmental qualities of Costa Rica and the international profiling of the country as an environmental champion. In fact, it is difficult to disentangle among these factors, but socio-environmental qualities did play a major role in the decision and were also prominent in the discourse of social movements opposing existing oil concessions. Echeverria (2016) discusses how the overlap between the 1997 Costa Rican oil concessions, indigenous territories, and national parks galvanized social resistance to oil operations (p. 11-12; 18). In particular, the movement *Acción de Lucha Anti-petrolera* was motivated primarily by local socio-environmental concerns and was also instrumental to the declaration of the 2002 moratorium (Echeverria, 2016 p. 32). Later moratoria were also accompanied by strong pressure by social movements as well as the recognition by President Solís of local environmental risks associated with oil operations (Echeverria, 2016 p. 33). For these reasons, we prefer to keep Costa Rica in the list.

- At lines 278-279, I think it is wrong to explain the USGS estimate being smaller than McGlade & Ekins' in terms of types of oil not included; a more important reason is that the USGS estimate is limited to resources with the potential to become reserves by 2025 plus undiscovered resources (see above), whereas McGlade & Ekins refer to all resources.

Addressed. Now it reads:

“There are significant differences between the georeferenced resources we use, and the non-georeferenced tabular data provided by other sources and datasets. We put together georeferenced data for 1,627 Gbbl of conventional oil resources, while ref.(Welsby et al., 2021) provided an estimate of 2,576 Gbbl of conventional oil resources. These differences are explained by the dearth of accessible and up-to-date spatial data.”

Furthermore, we also included the following sentence:

“The USGS WPA 2000 provides estimates of the quantities of conventional oil resources outside the United States that had the potential to be added to reserves from 1995 to 2025.”

- At lines 291-293, the statement that “our identification of exclusion zones and prioritization of unburnable oil resources is based not solely on socio-environmental criteria, but also economic ones” is overstated, as the economic optimisation relates only to the balance between oil, gas and coal, not to the prioritisation or exclusion within the oil portion.

Our analysis takes from Welsby et al 2021 the global total distribution of unburnable fuels among coal, gas and oil. We have now clarified this in the text:

“To identify the unburnable oil resources whose conservation would generate substantial collateral socio-environmental benefits, our spatial analysis builds upon the cost-optimal distribution of unburnable fuels among coal, gas and oil proposed by ref. Welsby et al. 2021 (i.e. a key assumption of our spatial analysis is that 71% of the conventional oil resources should remain unextracted by 2050, as well as 81% of conventional gas resources, 99% of unconventional oil resources, 93% of unconventional gas resources and 97% of coal -Supplementary Table 1). Although our identification of exclusion zones and prioritization of unburnable oil resources is based solely on socio-environmental criteria, it builds upon the supply costs for each fossil fuels resource category considered by ref. (Welsby et al. 2021).”

- At line 386, I would like to see literature references to support the use of a 10km radius around urban areas as “a safe zone considering health risks associated with oil extraction”.

Setting a safe distance to oil extraction regarding health risks is not an easy task. Health effects among people residentially exposed to upstream oil industry contaminants have been poorly studied, particularly in low- and middle-income countries (LMICs) (O’Callaghan et al 2016). In a systematic review, O’Callaghan et al 2016 only found 11 studies examining potential health effects of exposed communities in LMICs. 10 of these studies were conducted in the Ecuadorian and Peruvian Amazon and one in the Niger Delta. The development of unconventional oil and gas projects and the dramatic increase of fracking in the USA in the past decade started to create concern among environmental health researchers (Konkel, 2016) and

more environmental health studies are now available regarding fracking in high income countries (Johnston et al 2019, Deziel et al 2020). Besides the lack of scientific data, the use of sub-standard technologies for oil extraction in LMICs adds a further twist in the difficult task of setting a unique safe distance to oil extraction (Jernelöv, 2010). Similarly, the different environmental characteristics of the oil extraction locations in the globe render a uniform safe distance difficult. Globally, there are an estimated 70,000 oil fields across ~100 countries (Johnston et al 2019).

Regarding shale gas drilling, Texas (USA) permits drilling 200 ft (aprox. 60 m) from residences. However, many municipalities in Texas have established longer setback distances (setback distances from residences range from 300 to 1500 ft, aprox. 90 to 460 m) (Fry 2013). Setback distances in Texas have increased over time, and, according to Fry, rather than technically-based, setbacks are political compromises: “rigorous and empirical research was not utilized to determine or demarcate ‘safe’ or ‘healthy’ distances, and the setback distances are highly politicized compromises between residents’ concerns about the proximity of gas wells to their homes, mineral owners’ rights to profit from gas drilling, and the city council’s fear of legal lawsuits for a regulatory takings” (Fry 2013). In fact, McKenzie et al. (2012) found that residents living within 0.8 km from a fracking well are at a higher health risk than those farther away with benzene as the major contributor to the risk. Coons and Walker (2008) also found that significant ambient benzene emissions exist within close proximity to a fracking well (0.8 km), which resulted in significant public health problems. Methane concentrations in drinking water wells within 1 km of a fracking well can reach potentially explosive levels (Osborn et al., 2011). A recent working paper suggested that a distance within at least 2.5 km from a gas well is detrimental to fetus development due to the exposure to shale gas development (Hill, 2013). In the USA, a proximity within 1 km to a fracking well is a recurring critical value used for significant risks to environment and public health risk. Meng 2015 used the distances 1 km, 2 km, and 3 km as break values to group risk levels into high (≤ 1 km), moderate (1–2 km), and low (2–3 km) risks. Beyond 3 km, Meng 2015 assume there was little impact of fracking wells on the environment and inhabitants in USA. However, a much more recent review of the epidemiological research on unconventional oil and gas development and health outcomes (all of them conducted in USA or Australia) showed detectable health effects in distances of up to 10 miles (aprox. 16km) away (Deziel et al 2020).

In the very few studies that exist in LMICs, larger distances are suggested. O’Callaghan et al 2021, found that increased distance between residence to an oil processing facility (i.e. oil wells, central production facilities, gathering stations and pumping stations) in the Peruvian Amazon was associated with lower Blood Lead Levels (BLL) (O’Callaghan et al 2021). The median distance from the studied communities to a processing facility was 5.5km, and BLL decreased 5% every 10km up to 200km distance from the oil facilities. A study in south-eastern Bolivia that measured the levels of total petroleum hydrocarbons (TPH), polycyclic aromatic hydrocarbons (PAH), and 22 metals in the drinking water of residents living ≤ 30 km from an oil extraction field found that three-quarters of the samples were contaminated with concentrations exceeding the reference levels (Alonso et al., 2010). Similarly, a meta-analysis of chemical data from governmental institutions and oil companies reports proved that the dumping of produced water (i.e. the main waste product of oil extraction operations) increased lead, cadmium, chromium and barium concentrations in rivers of the Peruvian Amazon up to 36km downstream of the oil facilities (Yusta-Garcia et al., 2017).

Therefore, now the text reads:

“All urban areas, including a 10 km buffer around them, were designated as irreconcilable with oil extraction. The 10km buffer was set as a safe zone considering health risks associated with oil extraction.”

All that said, I do believe this is a very important piece of research, and I hope the authors are not disheartened by my more critical remarks. On the contrary, I would strongly encourage them to persevere.

Thank you for your encouraging words, we are happy to share a revised, and we believe much improved, version of the paper.

Reviewer #3 (Remarks to the Author):

This is a valuable contribution to the growing literature on fossil fuel supply-side approaches to climate mitigation efforts. The authors present findings that will be of significance to the academic and policy community, though they rightly note that more fine-grain analysis will be required to inform detailed policy and divestment/investment decisions. The work supports the claims provided, though it excludes from the onset unconventional oil reserves (as these are deemed unburnable in Ref. 1).

Thank you.

Major comments:

#3.1

The restriction considerations of Indigenous peoples to those “in voluntary isolation” is understandable, but it does not reflect the widespread and often unsuccessful resistance of other Indigenous peoples to fossil fuel development on their (unrecognized) territories. It was good to see this mentioned in the conclusion as a possible application of this method.

Thanks, a related point was made also by referee 2 and we address it in our reaction to comment #2.10 focusing on the issue of free, prior, informed consent.

#3.2

Two reserve categories are not given specific attention, and may require a very discussion or at least mention: Oil reserves in the Arctic and Ultra-deep sea reserves, given the concerns with major spills (e.g. Deepwater Horizon).

Thanks, we now included ‘further research’ options in the conclusions and the text reads:

“Atlases of gas and coal could be produced, and specific risks associated with extraction technologies in certain contexts (such as resources in the Arctic and ultra-deepwater resources) could also be used as exclusion criteria. Also, the use of more fine-grained data on reserves and specific reservoirs and oil fields could produce atlases of unburnable oil that would be appropriate for local and regional policymaking.”

#3.3

Regarding marine biodiversity protection, there is a mention on ln. 128 that “biodiversity hotspots are circumscribed to terrestrial areas”, which could be clarified as it may initially lead readers to think that marine biodiversity is not covered until later in the paper (through ref. 50 and the Marine Protected

Areas).

Thanks, we have now amended the text to clarify it:

"We use several indicators to capture the biological value of specific geographical areas: biodiversity hotspots⁴⁸, richness centres of terrestrial and marine endemic species^{49,50} and the global system of protected areas. The biodiversity hotspot approach⁴⁸ stands out as the best-known and the most widely accepted scheme to identify global biodiversity conservation priorities⁵¹. However, since biodiversity hotspots are circumscribed to terrestrial areas, we also considered the richness centres of terrestrial and marine endemic species in order to capture all global biodiversity conservation priorities."

#3.4

More generally, the analysis does not account for the relative risks of spills between reserve types, extraction technologies, and regions. This may be something that could be suggested among the list of 'further research' options.

We have now included this issue as a topic for future research, see point #3.2.

Finally, the exclusion of unconventional oil (Ln. 97-98) should perhaps be made clearer in the abstract and any communication, especially given the importance of unconventional reserves that are likely to continue being exploited (e.g. bitumen in Canada).

Addressed as suggested. The abstract now mentions "conventional" resources three times.

Minor comments:

Ln. 75-78, for impacts on fisheries and coastal communities, see for example: Andrews, N., Bennett, N. J., Le Billon, P., Green, S. J., Cisneros-Montemayor, A. M., Amongin, S., ... & Sumaila, U. R. (2021). Oil, fisheries and coastal communities: A review of impacts on the environment, livelihoods, space and governance. *Energy Research & Social Science*, 75, 102009.

Thanks for pointing out this important article. We now cite the suggested article in the manuscript when discussing the impacts of oil and gas extraction on fisheries and coastal ecosystems and communities.

Ln. 90, it is unclear if conventional oil includes unconventional oil (as mentioned Ln. 61) as unconventional oil is not mentioned in the subsequent sentence (only unburnable coal and natural gas). But this is almost immediately clarified at the beginning of the following paragraph, so probably no need to revise statement on Ln. 90.

We moved the text around to clarify:

“Here, we propose a methodology to identify and prioritize unburnable fossil fuel resources, apply it to the case of conventional oil and produce an atlas of unburnable conventional oil according to environmental and social criteria. The same methodology could be used to prepare atlases for unburnable coal and natural gas. We focus on conventional oil since almost all unconventional oil resources (1,518 Gbbl) should remain unburned because extraction costs are much higher (Welsby et al 2021).

To prioritize unburnable resources, we first assess their global distribution. Oil is usually categorized as ‘conventional’ or ‘unconventional’: oil with density lower than water (often standardized as ‘10° API’) is conventional (i.e., oil, light tight oil, and natural gas liquids -NGL-) and the remaining oil resources unconventional (i.e., natural bitumen, extra-heavy oil, and kerogen oil). Our georeferenced estimates of the different categories of conventional oil resources are presented in Figure 1.”

Ln. 395, minor typo (‘datalayer’).

Addressed as suggested.

Ln. 396 (and throughout the paper), the term Indigenous is capitalized by most Indigenous studies scholars, and people is generally used in the plural form (as done in 397).

Addressed as suggested.

Ln. 402, minor typo (‘.’).

Addressed as suggested.

REVIEWER COMMENTS

Reviewer #1 (Remarks to the Author):

Thank you for the opportunity to see the revised manuscript. In the main the authors have done a good job at addressing my comments. I have some more minor comments that the authors need to address prior to going forward to publication –

- Further motivation for this paper could also come from this new analysis - <https://secure.protected-carbon.org/>

- I think it is great that you have shifted the focus of the paper onto the more stringent climate target, based on Welsby et al. However, I had a couple of questions –

- o Lines 142-144: you state that “(for instance, estimates of unextractable conventional oil and gas resources are reduced to 54% and 76% respectively)⁹.” It is not clear to me where these values come from – or how they have been estimated.

- o Lines 220-222: you state that “Considering the carbon budget associated with the target of 1.5°C of global warming and applying a cost-optimal distribution of the carbon budget among coal, gas, and oil (following ref. Welsby et al. 2021), 71% (752 Gbbl) of the 1,823 Gbbl conventional oil resources can be extracted.” 71% should read 29%. Also, note the uncertainty around the 752 Gbbl value; as noted in Welsby, this value could be much lower if we were to rule out or reduce the role of CDR and use a more robust probability for 1.5C.

- I think you could make it clearer in the main paper the approach taken. In lines 231-232, you state that “To identify the unburnable conventional oil resources, we use biological and social criteria.” This should first state that you have already identified the total extractable (752) based on Welsby, and then you use these additional criteria to explore which areas should not be extracted from, based on social and environmental criteria. I think you just need to better bridge the information above and the text on socio-environmental criteria.

- Based on the new criteria you estimate that 447 Gbbl is excluded. So from your 1,627 Gbbl of conventional resource, this mean that the 752 extractable now needs to come from a reduced resource of 1180 Gbbl. I think this is a clearer way of expressing it than that put forward in lines 279-281 where you state “These 447 Gbbl are well below the 876 Gbbl of our georeferenced conventional oil resources required to be left unburned to keep global warming under 1.5°C.”.

- Line 356-358 is oddly expressed – “These zones overlap with 447 Gbbl of oil resources that by themselves are insufficient to meet climate policy targets, implying that they can be kept entirely off limits to contribute to climate policy and still other oil resources will also need to be left under the soil.” What do you mean ‘insufficient to meet climate policy’? I think you mean that ‘given that climate policy targets require that 876 Gbbl needs to stay in the ground, these additional criteria determine which of the available resource (1627 Gbbl) must remain in the ground. This helps shape the distribution of unextractable resource beyond economic factors’.
- In the conclusions, I think it would be good to reflect on how your method deals with trade-offs between economics and socio-environmental factors. I think if you had used a reserves basis (economic today based on available recovery methods), some of these trade-offs may have been easier to explore. I think this could be useful to consider in further research.
- Could you also reflect on how your method could deal with the unextractable share that has not been excluded (the difference between 876 and 447)? How should we think about that?
- Finally, in the conclusions note how a much smaller extractable estimate (given the uncertainty around 752 Gbbl) and more stringent socio-environmental considerations might impact the results.
- In lines 421-423, make it clear that current oil reserves is a subset of conventional and unconventional oil resources. Otherwise some might think that the values provided are all cumulative.

Reviewer #2 (Remarks to the Author):

Thank you to the authors for their further work on the paper, which as a result is much stronger. They have addressed most of my comments, especially with clarifying the timescales and using Welsby et al (2021) in place of McGlade & Ekins (2015).

I recommend this manuscript for publication, with some minor further revisions.

In some cases, the authors have helpfully explained why they could only partially address my comments or concerns. Sometimes I think they could include a bit more of this reasoning in the paper or Methods, including recognising the limitations of the approach and opportunities for future research.

For clarity, I will make further comments by relation to the numbering of my earlier comments; wherever one of my earlier comments is not referenced below, that is because I feel the authors have fully addressed it.

Before that, I have one new comment (sorry!), which might perhaps raise a more-than-minor issue.

Now that the geographical distribution of oil resources is more fleshed out in lines 182-188, I notice the striking absence of North America (other than Alaska) from the list of largest basins. Is this because tight oil was not considered extractable at the time of the USGS surveys? If so, this is a very significant limitation of the data source.

(I'm also a bit surprised that US and Canadian conventional (non-tight) basins are smaller than Gulf of Guinea, Brazil, Alaska or North Sea). Could this be that the basins are more tightly defined (hence smaller) in North America than elsewhere?

Perhaps there are other good explanations of why North American tight and non-tight oil are not among the largest (in which case, a note would be helpful).

2.3:

While the authors have improved the analysis with the addition of more up-to-date data, they still (by necessity) rely on the USGS studies from 2000 and 2012 for discovered resources. To this they add undiscovered amounts as estimated by studies after 2012.

This leads to an uneven set of moments in time at which the oil data are gathered, whereas to be really meaningful, a resource or reserve assessment needs to be specified at a specific point in time. It would be good to refer to the unextractable resources at date XXX.

Specifically, there are two time-related quantities that are not accounted for here:

- Amounts extracted since each assessment was made
- Amounts discovered after the 'discovered' assessment was made (2000 or 2012) but before the 'undiscovered' assessment was made.

Are the authors able to estimate the first quantity? Data are readily available on extraction by country; in some cases, national petroleum statistics can disaggregate extraction by basin, and where this is not available, perhaps an assumption could be made about proportionate split between basins/countries? Doing this would allow the data to be assigned to a chosen reference date (perhaps the reference date used in Welsby et al?), together with a caveat accounting for the second omission, such as:

"One weakness of this available data is that we omit discoveries made between YYY and ZZZ, hence somewhat understating the total unburnable oil associated with socio-environmental exclusion zones; we leave this to future research, when updated data on discovered resources become available."

Perhaps the authors could also comment on how much they would expect this omission to affect the results e.g. how prolific have been the discoveries in the most sensitive basins?

2.4:

The authors make a good case for why the data sources I suggested on reserves would not work for this study. They also explain/argue why resources is an instructive measure.

I suggest adding some of this explanation (e.g. the last 2 paragraphs of their response) to the Methods or Discussion.

One thing I still worry about is that not all resources can be considered economically extractable. So not all of the portion outside the exclusion zones can realistically be extracted. Therefore in the prioritising exercise on lines 313-322, I suggest adding a note that techno-economic factors will also have to be considered alongside socio-environmental factors in prioritising which resources remain

unextracted/unburned. (see ref 29, which aims to combine (different) equity and economic criteria in prioritising which to extract)

2.8:

All fair points, and good to have the extra text in Methods further explaining the sources. However, again, I would suggest explicitly noting the limitation of using urban areas as a proxy for population density.

2.9:

Just to check that I've interpreted correctly: since 60.2% of Zagros Fold overlaps with exclusion zones (for example), you judge 60.2% of the oil resources in that basin to be unburnable?

2.10

The added text on Indigenous Peoples at lines 371ff is good. I suggest adding that there is no global data available on Indigenous territories/FPIC, and this is an area for future research.

Small/specific issues:

- I still find the penultimate sentence of the abstract a bit unclear. I would find it clearer to say:

"The results show that biodiversity hotspots, richness centres of endemic species, natural protected areas, urban areas, and the territories of Indigenous peoples in voluntary isolation for 447 Gbbl of conventional oil resources. In identifying the 876 Gbbl of unburnable resources required to keep global warming under 1.5°C, these most socio-environmentally sensitive areas can be kept entirely off-limits to oil extraction."

- I appreciate the added text beginning at line 497. I suggest adding after the first sentence: “However, we believe the two are close enough in both definition and size to be a meaningful proxy” (perhaps also use the word proxy in the main text?).
- I found the reasoning and citations for selecting the 10km exclusion zone around urban areas very helpful. I suggest including some of this (e.g. page 20 of rebuttal letter) in Methods, or as Supplementary Text with a cross-reference from Methods.

Finally, some minor suggested edits to new text, for clarity purposes:

Lines 96-98: add “at current rates of emissions”

Line 98: I think you may have the wrong reference here – Huppman et al doesn’t make this point. (Also, I think the exhaustion year at current rates would be 2031?)

Line 104: suggest “associated risk of stranded assets”

Line 121: suggest “contribute towards realising developing countries’ right to develop” or (better) “contribute towards realising developing country populations’ right to socio-economic development”

Lines 122-128: this long sentence rather subsumes the key point of what the paper is doing. I suggest starting a new paragraph and breaking the sentence and making clearer the distinction:

"Previous studies based on ethical considerations suggest that the wealthiest countries should leave their fossil fuel reserves underground, whereas economic efficiency arguments suggest that the costliest resources should be left untapped. In this paper, we take a different approach: We propose that the socio-environmental impacts of extraction in different locations gives an alternative, ethics-based distribution of which fossil fuels are unburnable."

Lines 135 and 143: I'm not clear where these %ages of conventional oil and gas come from in Welsby et al. But Supp Table 1 in that paper has 71% and 81% of all (not just conventional) oil and gas unextractable by 2100 (not 2050), and I think this is a more interesting number anyway than the %age of conventional by 2050.

Lines 140-144: I think this sentence is overly complicated and easily misread. If you agree with me on the previous point, I suggest just deleting this sentence.

Line 176: I would suggest a brief note highlighting that the definition of "conventional" includes tight oil, whereas often fracking is considered unconventional.

Lines 380-382: I'm not convinced this helps investors minimise stranded asset risk; their use of this analysis would only help them avoid stranded assets if regulators were to adopt and enforce the same exclusions. Stranded assets are appropriately mentioned in the introduction; I suggest deleting this sentence from the conclusion.

Reviewer #3 (Remarks to the Author):

This is an excellent revision.

p. 1, Typo: 11,000 GtCO₂

p.4, ln. 108-109, A recent study assessed the determinants of adoption of supply-side policies by governments,

Lujala, P., Le Billon, P., & Gaulin, N. (2022). Phasing out fossil fuels: Determinants of production cuts and implications for an international agreement. *Global Environmental Politics*, 22(4), 95-128.

p. 7, ln. 173, It may be worth mentioning that unconventional oil resources also tend to have a lower energy efficiency (ERoEI/EROI) and higher environmental impacts. In addition, use reference number rather than (name, year) format.

p. 8/p. 24, Consider explaining, either in the main text or the Methods section, that oil is assumed to be uniformly distributed within each sedimentary basin.

p. 11, ln. 239-242, consider adding a source and/or clarifying the statement.

p. 16, Table 2, Consider checking again the characteristics of oil taken into consider by the study for Canada, as most of its resources fall under the unconventional oil resources, not conventional oil resources. As such the percentage of Regional unburnable conventional oil (%) may be much higher (lower denominator, which here could be unconventional oil resources rather than conventional ones). The same may apply for Central and South America because of Venezuela's large unconventional oil resources.

p. 20, ln. 368, Correct typo "Ar[c]tic"

p. 28, ln. 518, Mention "natural gas"

REVIEWER COMMENTS

Reviewer #1 (Remarks to the Author):

Thank you for the opportunity to see the revised manuscript. In the main the authors have done a good job at addressing my comments. I have some more minor comments that the authors need to address prior to going forward to publication –

1.1. Further motivation for this paper could also come from this new analysis - <https://secure.protected-carbon.org/>

Thanks for bringing this project to our attention. The initiative stresses the potential of protected areas to contribute to climate policy by limiting the extraction of fossil fuels and is within the context of two broader but very pertinent research initiatives that have gained traction during the review process of the manuscript Leave It in the Ground (LINGO) and Oil Change International. We now have included a reference to these initiatives in the main text.

1.2. Lines 142-144: you state that “(for instance, estimates of unextractable conventional oil and gas resources are reduced to 54% and 76% respectively)⁹.” It is not clear to me where these values come from – or how they have been estimated.

The text now clarifies the source of the figures, and mentions CDR and the probabilities to achieve climate objectives, also addressing the next comment (1.3):

‘According to Ref.⁸, estimates of unextractable conventional oil and gas resources by 2100 are reduced to 54% and 76% respectively.⁸ It is important to note that continued use of fossil fuels after 2050 depends also on Carbon Dioxide Removal (CDR) and, more generally, that the proportions of extractable oil depends on the carbon budget, which in turn is a function of the desired probability of achieving climate objectives. More certainty would require more fossil fuels to stay underground.’

1.3. Lines 220-222: you state that “Considering the carbon budget associated with the target of 1.5°C of global warming and applying a cost-optimal distribution of the carbon budget among coal, gas, and oil (following ref. Welsby et al. 2021), 71% (752 Gbbl) of the 1,823 Gbbl conventional oil resources can be extracted.” 71% should read 29%. Also, note the uncertainty around the 752 Gbbl value; as noted in Welsby, this value could be much lower if we were to rule out or reduce the role of CDR and use a more robust probability for 1.5C.

Thanks for flagging this, the text now reads 29%.

1.4. I think you could make it clearer in the main paper the approach taken. In lines 231-232, you state that “To identify the unburnable conventional oil resources, we use biological and social criteria.” This should first state that you have already identified the total extractable (752) based on Welsby, and then you use these additional criteria to explore which areas should not be extracted from, based on social and environmental criteria. I think you just need to better bridge the information above and the text on socio-environmental criteria.

The text now reads: “To identify these 1,524 Gbbl of unburnable conventional oil resources, we use biological and social criteria.”

(Consider that all the figures have changed as we have now included, following the comments of reviewer 2, global resources on Light Tight Oil –LTO-).

1.5. Based on the new criteria you estimate that 447 Gbbl is excluded. So from your 1,627 Gbbl of conventional resource, this mean that the 752 extractable now needs to come from a reduced resource of 1180 Gbbl. I think this is a clearer way of expressing it than that put forward in lines 279-281 where you state “These 447 Gbbl are well below the 876 Gbbl of our georeferenced conventional oil resources required to be left unburned to keep global warming under 1.5°C.”.

Throughout the section we compare the exclusion and then the priority criteria to the amount of identified unburnable conventional oil as a portion of conventional oil resources, and we would find it disorienting to turn things around and introduce the budget of 1,181 Gbbl. Just to be clear, the methods section includes the sentence:

‘Out of the 2,276 Gbbl of georeferenced conventional oil resources, 1,524 Gbbl (2,276 minus 752 Gbbl) should be left untapped.’

1.6. Line 356-358 is oddly expressed – “These zones overlap with 447 Gbbl of oil resources that by themselves are insufficient to meet climate policy targets, implying that they can be kept entirely off limits to contribute to climate policy and still other oil resources will also need to be left under the soil.” What do you mean ‘insufficient to meet climate policy’? I think you mean that ‘given that climate policy targets require that 876 Gbbl needs to stay in the ground, these additional criteria determine which of the available resource (1627 Gbbl) must remain in the ground. This helps shape the distribution of unextractable resource beyond economic factors’.

The sentence now reads:

‘These zones overlap with 609 Gbbl of oil resources, while keeping global warming under 1.5°C requires that 1,524 Gbbl stay in the ground. Thus, oil resources in exclusion zones can be kept entirely untapped and still additional oil resources will also need to be left under the soil.’

1.7. In the conclusions, I think it would be good to reflect on how your method deals with trade-offs between economics and socio-environmental factors. I think if you had used a reserves basis (economic today based on available recovery methods), some of these trade-offs may have been easier to explore. I think this could be useful to consider in further research.

We now include two sentences about future research.

In the conclusions: 'Additional research could also introduce ways to negotiate tradeoffs between different criteria.'

And in the 'Methods': " Nevertheless, we acknowledge that an analysis of reserves is an important next step in the research on unextractable fossil fuels and our methodology could be adapted to construct atlases for unburnable fossil fuel reserves".

1.8. Could you also reflect on how your method could deal with the unextractable share that has not been excluded (the difference between 876 and 447)? How should we think about that?

Yes, that would be possible through ranking:

'we created rankings to achieve the conservation of additional oil resources to reach a total of 1,524 Gbbl of unburnable oil. The ranking prioritizes resources according to terrestrial and marine biodiversity and human population densities.'

1.9. Finally, in the conclusions note how a much smaller extractable estimate (given the uncertainty around 752 Gbbl) and more stringent socio-environmental considerations might impact the results.

In the reaction to the first comment, we have already introduced a sentence about the effect of increased certainty over the carbon budget and, as a consequence, over the share of unburnable fossil fuels. Please see above (1.2).

- In lines 421-423, make it clear that current oil reserves is a subset of conventional and unconventional oil resources. Otherwise some might think that the values provided are all cumulative.

The text includes the sentence:

'Reserves are a subset of oil resources'

Reviewer #2 (Remarks to the Author):

2.1. Thank you to the authors for their further work on the paper, which as a result is much stronger. They have addressed most of my comments, especially with clarifying the timescales and using Welsby et al (2021) in place of McGlade & Ekins (2015).

I recommend this manuscript for publication, with some minor further revisions.

Thank you.

In some cases, the authors have helpfully explained why they could only partially address my comments or concerns. Sometimes I think they could include a bit more of this reasoning in the paper or Methods, including recognising the limitations of the approach and opportunities for future research.

For clarity, I will make further comments by relation to the numbering of my earlier comments; wherever one of my earlier comments is not referenced below, that is because I feel the authors have fully addressed it.

2.2. Before that, I have one new comment (sorry!), which might perhaps raise a more-than-minor issue.

Now that the geographical distribution of oil resources is more fleshed out in lines 182-188, I notice the striking absence of North America (other than Alaska) from the list of largest basins. Is this because tight oil was not considered extractable at the time of the USGS surveys? If so, this is a very significant limitation of the data source.

(I'm also a bit surprised that US and Canadian conventional (non-tight) basins are smaller than Gulf of Guinea, Brazil, Alaska or North Sea). Could this be that the basins are more tightly defined (hence smaller) in North America than elsewhere?

Perhaps there are other good explanations of why North American tight and non-tight oil are not among the largest (in which case, a note would be helpful).

We thank the reviewer for raising this important issue.

There is no full consensus on whether Light Tight Oil (LTO) should be considered conventional or unconventional oil and some of the world energy standard setting institutions (i.e. USGS, SPE, EIA and IEA) define LTO as unconventional⁵⁹⁻⁶². The spatial analysis of our manuscript was based on a spatial dataset of global *conventional* oil resources from the USGS and, therefore, did not include data on LTO. However, we note that a) Welsby et al. 2021 do include LTO as conventional oil as they define conventional oil as oil having an API index greater than 10° and, b) our spatial prioritization of unburnable *conventional* oil builds

upon the cost-optimal distribution of unburnable fuels among the different categories calculated by Welsby et al. 2021 considering LTO as conventional oil. To avoid this inconsistency, we have now included in the analysis the available open-sourced global spatial estimates of LTO resources: those provided by EIA (the US Energy Information Administration). For USA LTO resources, we sourced data from the report 'Assumptions to the Annual Energy Outlook 2023: Oil and Gas Supply Module' (https://www.eia.gov/outlooks/aeo/assumptions/pdf/OGSM_Assumptions.pdf), which provide estimates from 2021. For world LTO (excluding the USA), estimates were retrieved from the report 'Technically Recoverable Shale Oil and Shale Gas Resources: An Assessment of 137 Shale Formations in 41 Countries Outside the United States' (https://www.eia.gov/analysis/studies/worldshalegas/archive/2013/pdf/fullreport_2013.pdf?zscb=97852318), updated in December 2014 (https://www.eia.gov/analysis/studies/worldshalegas/xls/Attachments_A_B_by_Country_ARI_EIA_World_Shale_Resources_122914.xlsx).

Considering this new data, now US ranks second in the list of largest basins on conventional oil resources (including LTO). The text now reads:

“Our georeferenced estimates of conventional oil resources are presented in Figure 1. They show that the global spatial distribution of conventional oil resources is uneven and, out of the total of 2,276 Gbbl, the largest ones are in the sedimentary basins of Middle East (648 Gbbl or 28%; mainly in the Mesopotamian Foredeep Basin, 267 Gbbl, Zagros Fold Belt, 117 Gbbl, Greater Ghawar Uplift, 96 Gbbl and Rub Al Khali Basin, 101 Gbbl), the United States (402 Gbbl or 18%; mainly in the Permian Basin, 181 Gbbl, Gulf Coast Basins, 64 Gbbl, Northern Alaska, 36 Gbbl and, Appalachian Basin, 34 Gbbl), Russia and former Soviet states (343 Gbbl, or 15%; mainly in the West Siberia Basin, 189 Gbbl), Gulf of Guinea (West Central Coast, 61 Gbbl, and Niger Delta, 43 Gbbl), off-shore Brazil (Santos basin, 66 Gbbl, and Campos Basin, 24 Gbbl), North Africa (Sirte Basin, 46 Gbbl, Trias/Ghadames Basin, 24 Gbbl) and the North Sea (33 Gbbl)”.

We have also clarified in the manuscript the use of LTO. The text now reads:

“We follow Ref.⁸ that defines oil with density lower than water (often standardized as '10° API') as conventional (i.e., oil, light tight oil –LTO–, and natural gas liquids -NGL) and the remaining oil resources as unconventional (i.e., natural bitumen, extra-heavy oil, and kerogen oil). However, it is important to note that the world energy institutions (i.e. USGS, SPE, EIA and IEA) define LTO as unconventional –since LTO does not flow without stimulation and require specialized extraction technology (i.e. hydraulic fracturing).”

...

The reviewer is also 'a bit surprised that US and Canadian conventional (non-tight) basins are smaller than Gulf of Guinea, Brazil, Alaska or North Sea'. This is partially explained, as assumed by the reviewer, because 'the basins are more tightly defined (hence smaller) in North America than elsewhere'. We identified some of the largest basins (we did not present the data aggregated by geographical regions), indicating their geographical regions to make it easier to understand for the reader. However, as expected by the reviewer, the aggregated USA (including Alaska) and Canadian basins contain 87.1Gbbl (65.5+21.6Gbbl) (excluding LTO), exceeding the conventional oil resources of the Santos basin, 66 Gbbl (off-shore Brazil), the West Central Coast basin, 61 Gbbl (West Central Coast of Africa), those of the Gulf of Guinea (Niger Delta basin, 43 Gbbl and, Gulf of Guinea basin 5Gbbl) and, the North Sea (33 Gbbl).

2.3. While the authors have improved the analysis with the addition of more up-to-date data, they still (by necessity) rely on the USGS studies from 2000 and 2012 for discovered resources. To this they add undiscovered amounts as estimated by studies after 2012.

This leads to an uneven set of moments in time at which the oil data are gathered, whereas to be really meaningful, a resource or reserve assessment needs to be specified at a specific point in time. It would be good to refer to the unextractable resources at date XXX.

Specifically, there are two time-related quantities that are not accounted for here:

- Amounts extracted since each assessment was made
- Amounts discovered after the 'discovered' assessment was made (2000 or 2012) but before the 'undiscovered' assessment was made.

Are the authors able to estimate the first quantity? Data are readily available on extraction by country; in some cases, national petroleum statistics can disaggregate extraction by basin, and where this is not available, perhaps an assumption could be made about proportionate split between basins/countries? Doing this would allow the data to be assigned to a chosen reference date (perhaps the reference date used in Welsby et al?), together with a caveat accounting for the second omission, such as:

"One weakness of this available data is that we omit discoveries made between YYY and ZZZ, hence somewhat understating the total unburnable oil associated with socio-environmental exclusion zones; we leave this to future research, when updated data on discovered resources become available."

Perhaps the authors could also comment on how much they would expect this omission to affect the results e.g. how prolific have been the discoveries in the most sensitive basins?

We thank the reviewer for pointing out these important issues. We do agree with the reviewer that the main data limitations are the lack of data on: 1) extracted conventional oil (or 'oil production') since the USGS assessments were made (WPA 2000) and, 2) the volumes discovered (volumes added to prospective/discovered resources from prospective/undiscovered resources –according to the definition of SPE⁶²-) between the WPA 2000 and the WPA 2012 or the individual USGS assessments published for each particular sedimentary basin from 2012 onwards. As described in the methods section, we calculated conventional oil resources volumes by adding remaining discovered conventional oil resources as in the WPA 2000 to, either the undiscovered conventional oil resources as in the WPA 2012 or the individual USGS assessments for each sedimentary basin published from 2012 onwards.

Although we acknowledge the value added of estimating the first quantity (1), as proposed by the reviewer, we rather stick to our data for the following reasons:

To our knowledge, there is no data on oil extraction by country disaggregated by basin. Proportionally assigning the oil extracted in each country to the different national sedimentary basins according either to their surface area or their remaining resources would distort the data and create an artefact with implications that are difficult to gauge, ultimately undermining the benefit of having more updated data. In fact, several countries contain many basins (e.g. 67 in the case of the USA), some basins overlap with different countries and, we do not have information to help us to assess the development potential of the

remaining resources in each basin. Overall, we prefer presenting less updated but also less distorted data and acknowledging the limitation. Moreover, we have now quantified total global oil extraction since 2000 in order to clarify the limitations of our data. We include this data, disaggregated per country, in the Supplementary Information (Supplementary Table 3).

The text now reads:

“One limitation of our data is that we do not account for (a) conventional oil extraction between 2000 and 2022 (2013-2022 for USA) and, (b) discoveries made between 2000 and 2012+ (i.e. resources added to contingent /discovered resources from prospective/undiscovered resources –according to the definition of SPE- between the WPA 2000 and the WPA 2012 or the individual USGS assessments published for each particular sedimentary basin from 2012 onwards). Although there are no available data on the conventional oil extracted in each basin between 2000 and 2022, 751 Gbbl of conventional oil have been extracted globally in this period (IEA 2023) -see Supplementary Table 3 in the Supplementary Information. While this omission tends to overestimate the amount existing resources, the additions to the contingent resources category from the prospective resources category between 2000 and 2012+ produces an underestimation. It is worth mentioning that between 2000 and 2022, global reserves have increased by 685 Gbbl -585Gbbl between 2000 and 2012- (IEA 2023). Future research, based on up-to-date data on oil extraction and discovered resources at the basin level, could gauge the implications of updated oil resource estimates for the spatial distribution of unburnable oil.”

2.5. The authors make a good case for why the data sources I suggested on reserves would not work for this study. They also explain/argue why resources is an instructive measure.

I suggest adding some of this explanation (e.g. the last 2 paragraphs of their response) to the Methods or Discussion.

We have now added the following text in the *Methods* section:

“Although reserves are more likely to be extracted than resources, our analysis is focused on resources for several reasons. From a policy perspective, an analysis of resources is a first logical step towards the fulfilment of the Paris Agreement commitments since an analysis of resources can guide investment in oil exploration. At the same time, reserve volume estimates fluctuate over time depending on prices, the cost of available technologies, the development of new oil extraction technologies, new discoveries, and strategic overestimation by rights holders^{74,75}. There is much uncertainty on future energy prices which heavily depend on some key political choices, including the climate mitigation policies adopted by governments around the world. Ultimately, reserves depend crucially on climate policies, turning an analysis focusing on reserves into a questionable policy tool. Nevertheless, we acknowledge that an analysis of reserves is an important next step in the research on unextractable fossil fuels and our methodology could be adapted to construct atlases for unburnable fossil fuel reserves”.

One thing I still worry about is that not all resources can be considered economically extractable. So not all of the portion outside the exclusion zones can realistically be extracted. Therefore in the prioritising exercise on lines 313-322, I suggest adding a note that techno-economic factors will also have to be considered alongside socio-environmental factors in prioritising which resources remain

unextracted/unburned. (see ref 29, which aims to combine (different) equity and economic criteria in prioritising which to extract)

We have now added the following text that complements the response we gave also to comment 1.7 above:

“For example, future research could combine techno-economic factors alongside socio-environmental criteria to identify additional unburnable resources and include oil prices and production costs in the analysis. This is particularly salient, since we are using data on ‘technically recoverable resources’ (as opposed to ‘economically recoverable resources’) and this category includes all the oil that can be produced based on current technology, industry practice, and geologic knowledge, regardless of their economic feasibility. Oil prices and production costs specific to each basin could be included in the analysis and multiple techno-economic factors combined alongside socio-environmental criteria to identify unburnable resources³²”.

2.8. All fair points, and good to have the extra text in Methods further explaining the sources. However, again, I would suggest explicitly noting the limitation of using urban areas as a proxy for population density.

The text now reads:

“Although these ‘urban areas’ also include industrial areas (i.e areas that could arguably be suitable for oil extraction), they are, to our knowledge, the best proxy for global data on densely populated areas (both rural and urban)”.

2.9. Just to check that I’ve interpreted correctly: since 60.2% of Zagros Fold overlaps with exclusion zones (for example), you judge 60.2% of the oil resources in that basin to be unburnable?

The reviewer's interpretation is correct. Since more fine-grain geographic information (i.e. oil fields or reservoirs) on the location of the resources is not available, we assumed a homogeneous distribution of conventional oil resources throughout the oil basins. Conventional oil resources present at each geologic assessment unit (i.e. basin) were divided and proportionally assigned to each pixel of the geologic assessment unit according to its surface area. Thus, if x% of the basin coincides with exclusion zones, we consider x% of resources to be unburnable.

The lack of access to spatial data on oil resources at a higher resolution is the main data limitation of the present work, as we acknowledge in the manuscript.

2.10. The added text on Indigenous Peoples at lines 371ff is good. I suggest adding that there is no global data available on Indigenous territories/FPIC, and this is an area for future research.

Thank you. We have now added this in the main text:

“While there are no global data on this criterion available at this moment, this could be an important area for future research.”

Small/specific issues:

- I still find the penultimate sentence of the abstract a bit unclear. I would find it clearer to say:

"The results show that biodiversity hotspots, richness centres of endemic species, natural protected areas, urban areas, and the territories of Indigenous peoples in voluntary isolation for 447 Gbbl of conventional oil resources. In identifying the 876 Gbbl of unburnable resources required to keep global warming under 1.5°C, these most socio-environmentally sensitive areas can be kept entirely off-limits to oil extraction."

Thank you for the suggestion. We have rewritten the sentence following the reviewer's indications. Now the abstract reads:

"The results show that biodiversity hotspots, richness centres of endemic species, natural protected areas, urban areas, and the territories of Indigenous peoples in voluntary isolation coincide with 609 Gbbl of conventional oil resources. Since, 1,524 Gbbl of conventional oil resources are required to be left untapped in order to keep global warming under 1.5°C, all the above mentioned socio-environmentally sensitive areas can be kept entirely off-limits to oil extraction."

- I appreciate the added text beginning at line 497. I suggest adding after the first sentence: "However, we believe the two are close enough in both definition and size to be a meaningful proxy" (perhaps also use the word proxy in the main text?).

Addressed as suggested.

- I found the reasoning and citations for selecting the 10km exclusion zone around urban areas very helpful. I suggest including some of this (e.g. page 20 of rebuttal letter) in Methods, or as Supplementary Text with a cross-reference from Methods.

Addressed as suggested. We have now added the text in the 'Supplementary Information' file with a cross-reference in the 'Methods' section.

Lines 96-98: add "at current rates of emissions"

Addressed as suggested.

Line 98: I think you may have the wrong reference here – Huppman et al doesn't make this point. (Also, I think the exhaustion year at current rates would be 2031?)

We have changed the reference for:

IPCC, 2023: Climate Change 2023: Synthesis Report. Contribution of Working Groups I, II and III to the Sixth Assessment Report of the Intergovernmental Panel on Climate Change [Core Writing Team, H. Lee and J. Romero (eds.)]. IPCC, Geneva, Switzerland, 184 pp., doi: 10.59327/IPCC/AR6-9789291691647.

Line 104: suggest “associated risk of stranded assets”

Addressed as suggested.

Line 121: suggest “contribute towards realising developing countries’ right to develop” or (better) “contribute towards realising developing country populations’ right to socio-economic development”

In the text we were referring specifically to the right to development stated in the United Nations Declaration on the Right to Development of 1986, and in the Vienna Declaration of 1993. Now we adopted the very same language of the declarations, referring to the right to development (as opposed to the right to develop) and the reference at the end of the sentence explains at length the meaning and ethics of this right.

Lines 122-128: this long sentence rather subsumes the key point of what the paper is doing. I suggest starting a new paragraph and breaking the sentence and making clearer the distinction:

"Previous studies based on ethical considerations suggest that the wealthiest countries should leave their fossil fuel reserves underground, whereas economic efficiency arguments suggest that the costliest resources should be left untapped. In this paper, we take a different approach: We propose that the socio-environmental impacts of extraction in different locations gives an alternative, ethics-based distribution of which fossil fuels are unburnable."

Addressed. Now it reads:

“Previous studies suggest, based on ethical considerations, that developed countries are those who should leave their fossil fuels underground, whereas studies based on economic efficiency suggest that those resources whose extraction generates lower rents (because of relatively higher extraction, transportation and transformation costs, or lower economic value of the resources) should be left untapped. In this paper, we take a different approach and we use the socio-environmental impacts of fossil fuel extraction in different locations to suggest alternative distributions of unburnable fossil fuels.”

Lines 135 and 143: I’m not clear where these %ages of conventional oil and gas come from in Welsby et al. But Supp Table 1 in that paper has 71% and 81% of all (not just conventional) oil and gas unextractable by 2100 (not 2050), and I think this is a more interesting number anyway than the %age of conventional by 2050.

We built *Supplementary Table 1* based on data by 2050 from ref.⁸. In particular, from data available in the files:

- ‘Supplementary Information’:
https://static-content.springer.com/esm/art%3A10.1038%2Fs41586-021-03821-8/MediaObjects/41586_2021_3821_MOESM1_ESM.pdf
- ‘Source data for the Supplementary Figures’:

https://static-content.springer.com/esm/art%3A10.1038%2Fs41586-021-03821-8/MediaObjects/41586_2021_3821_MOESM3_ESM.xlsx

We also requested and obtained some extra data from the authors.

We run our spatial analysis using the %ages of unburnable conventional oil resources calculated by ref.⁸ by 2050. We use these %ages instead of the ones Welsby et al. calculated by 2100 in order to limit the substantial uncertainties around the scaling of CDR.

Lines 140-144: I think this sentence is overly complicated and easily misread. If you agree with me on the previous point, I suggest just deleting

We simplified the sentence and addressed also comment 1.2 It now reads:

“According to Ref. ⁸, estimates of unextractable conventional oil and gas resources by 2100 are reduced to 54% and 76% respectively.⁸ It is important to note that continued use of fossil fuels after 2050 depends also on Carbon Dioxide Removal (CDR) and, more generally, that the proportions of extractable oil depends on the carbon budget, which in turn is a function of the desired probability of achieving climate objectives. More certainty would require more fossil fuels to stay underground.”

Line 176: I would suggest a brief note highlighting that the definition of “conventional” includes tight oil, whereas often fracking is considered unconventional.

Addressed as suggested and also in response to comment 2.2.

Lines 380-382: I’m not convinced this helps investors minimise stranded asset risk; their use of this analysis would only help them avoid stranded assets if regulators were to adopt and enforce the same exclusions. Stranded assets are appropriately mentioned in the introduction; I suggest deleting this sentence from the conclusion.

We do believe that, by guiding future investments, the atlas of unburnable fuels can reduce the risk of stranded assets. The risk of stranded fossil fuel assets is due to the discrepancy between the carbon budget and the CO² emissions embedded in global fossil fuel resources . Not investing in the exploration and development of those resources that have a major risk to be subject to environmental policies (beyond climate policies), can prevent energy corporations, investors and national governments to invest in assets at risk of stranding. The atlas, identifies fossil fuel resources that overlap socio-environmental sensitive areas and, therefore, have more possibilities to be protected by future climate and environmental policies as well as be target of social movements e.g. divestment campaigns⁷² . We have amended the text to make it more explicit:

“It also makes it possible for energy corporations, governments and, more generally, investors to minimize the risks of stranded assets by highlighting those fossil fuel resources that overlap socio-environmental sensitive areas and, therefore, reduce the possibility of being impacted by future environmental policies (including and beyond climate policies) or becoming the target of contentious actions by social movements –e.g. divestment campaigns⁷²” .

Reviewer #3 (Remarks to the Author):

This is an excellent revision.

Thank you very much, we adopted all the following suggestions.

p. 1, Typo: 11,000 GtCO₂

p.4, ln. 108-109, A recent study assessed the determinants of adoption of supply-side policies by governments,

Lujala, P., Le Billon, P., & Gaulin, N. (2022). Phasing out fossil fuels: Determinants of production cuts and implications for an international agreement. *Global Environmental Politics*, 22(4), 95-128.

p. 7, ln. 173, It may be worth mentioning that unconventional oil resources also tend to have a lower energy efficiency (ERoEI/EROI) and higher environmental impacts. In addition, use reference number rather than (name, year) format.

p. 8/p. 24, Consider explaining, either in the main text or the Methods section, that oil is assumed to be uniformly distributed within each sedimentary basin.

p. 11, ln. 239-242, consider adding a source and/or clarifying the statement.

p. 16, Table 2, Consider checking again the characteristics of oil taken into consider by the study for Canada, as most of its resources fall under the unconventional oil resources, not conventional oil resources. As such the percentage of Regional unburnable conventional oil (%) may be much higher (lower denominator, which here could be unconventional oil resources rather than conventional ones). The same may apply for Central and South America because of Venezuela's large unconventional oil resources.

p. 20, ln. 368, Correct typo "Ar[c]tic"

p. 28, ln. 518, Mention "natural gas"

REVIEWER COMMENTS

Reviewer #1 (Remarks to the Author):

Thank you for the opportunity to see this revision of the manuscript. The authors have done a great job at addressing mine (and others) comments. I have some more minor comments but overall I would recommend this paper for publication. The number listed refer to my specific comments in the last round.

1.1 I am not sure I would call OCI an initiative per se. They are a campaign organization.

1.2 You add the following – ‘It is important to note that continued use of fossil fuels after 2050 depends also on Carbon Dioxide Removal (CDR) and, more generally, that the proportions of extractable oil depends on the carbon budget, which in turn is a function of the desired probability of achieving climate objectives. More certainty would require more fossil fuels to stay underground.’ I am not sure you need to expand the point so much; I would focus on the point of CDR; a reader might expect net-zero after 2050 – hence the importance of mentioning CDR, which in effect increases the available carbon budget for fossil fuels. I’d drop text from ‘.....more generally,.....’ onwards.

1.3 Ok.

The text between 165-170 should be clearer. You include two totals for conventional oil resources - 2,575 (Welsby) and your own 2,276.

One question is why you do not apply the Welsby percentage of 29% to your own estimate to get an extractable amount? I think you should take some time to explain this carefully in the Methods, including that you use the Welsby extractable amount and subtract this from your own estimate to determine the unextractable level. In the main text, be clearer as to what you mean by ‘adapt’ to get from one total estimate to the other. Otherwise it is confusing.

1.4 I am not sure your proposed edit is sufficient. What you want to say (I think) is that from the unextractable figure you have derived from Welsby, you want to further characterise what can’t be extracted based on social and environmental criteria – as additional factors for non-extraction. Please elaborate some more on this point to make the bridge between sections.

1.5 OK, my misunderstanding here.

But I might recommend that in lines 216-219, you discuss the resources subject to exclusion based on socio-environmental criteria as percentage of unextractable resources, not total resources (as currently). This feels more logical, due to this section being about the unextractable portion to which you then apply these additional criteria.

1.6 This seems good to me

1.7 Ok

1.8 Ok

1.9 Ok

Reviewer #2 (Remarks to the Author):

Thanks to the authors for answering my comments and making revisions where appropriate. I believe my comments have been fully addressed, and am happy to recommend the article for publication. Congratulations on an important piece of research

Response to the referees' comments

Ms. Ref. No.: NCOMMS-21-41044B

Title: The atlas of unburnable oil for supply-side climate policies

We would like to thank the referees for the encouraging assessment of our manuscript and for the comments that they have provided.

Reviewers' comments:

Reviewer #1:

Thank you for the opportunity to see this revision of the manuscript. The authors have done a great job at addressing mine (and others) comments. I have some more minor comments but overall I would recommend this paper for publication. The number listed refer to my specific comments in the last round.

Thank you.

1.1 I am not sure I would call OCI an initiative per se. They are a campaign organization.

We now refer to 'Initiatives and campaigns' in the text.

1.2 You add the following – 'It is important to note that continued use of fossil fuels after 2050 depends also on Carbon Dioxide Removal (CDR) and, more generally, that the proportions of extractable oil depends on the carbon budget, which in turn is a function of the desired probability of achieving climate objectives. More certainty would require more fossil fuels to stay underground.' I am not sure you need to expand the point so much; I would focus on the point of CDR; a reader might expect net-zero after 2050 – hence the importance of mentioning CDR, which in effect increases the available carbon budget for fossil fuels. I'd drop text from '.....more generally,.....' onwards.

We followed the advice and deleted the redundant text.

1.3 Ok.

The text between 165-170 should be clearer. You include two totals for conventional oil resources - 2,575 (Welsby) and your own 2,276.

One question is why you do not apply the Welsby percentage of 29% to your own estimate to get an extractable amount? I think you should take some time to explain this carefully in the Methods, including that you use the Welsby extractable amount and subtract this from your own estimate to

determine the unextractable level. In the main text, be clearer as to what you mean by 'adapt' to get from one total estimate to the other. Otherwise it is confusing.

In the methods section, we include:

"We put together georeferenced data for 2,276 Gbbl of conventional oil resources, while ref.8 provided a global estimate of 2,575 Gbbl. This difference is explained by the dearth of accessible and up-to-date spatial data. Ref.8 estimated that, based on the remaining carbon budget, 752 Gbbl of conventional oil resources could be burned. We subtracted this amount from our own resource estimates to determine the amount of conventional oil resources that cannot not be extracted. Thus, out of the 2,276 Gbbl of georeferenced conventional oil resources, 1,524 Gbbl (2,276 minus 752 Gbbl) should be left untapped and we use socio-environmental criteria to rank and prioritize these resources."

In the main text, the word 'adapt' should have read 'adopt'. We think that now the text is clear also in relation to the next comment.

1.4 I am not sure your proposed edit is sufficient. What you want to say (I think) is that from the unextractable figure you have derived from Welsby, you want to further characterise what can't be extracted based on social and environmental criteria – as additional factors for non-extraction. Please elaborate some more on this point to make the bridge between sections.

Because of the typo ('adapt' for 'adopt'), the two paragraphs were not properly bridged. We now conclude the paragraph with:

'We adopt this estimate of extractable oil and out of our 2,276 Gbbl of conventional oil resources we consider that 1,524 Gbbl need to be left untapped (see the *Methods* section for a detailed explanation).'

And start the next one with:

'To identify these 1,524 Gbbl of unburnable conventional oil resources, we use biological and social criteria.'

Thus, first, we explain how we derive the 1,524 Gbbl of unburnable resources and then refer to the figure and explain how we prioritized the unburnable resources –we believe the passage is very clear now.

1.5 OK, my misunderstanding here.

But I might recommend that in lines 216-219, you discuss the resources subject to exclusion based on socio-environmental criteria as percentage of unextractable resources, not total resources (as currently). This feels more logical, due to this section being about the unextractable portion to which you then apply these additional criteria.

Addressed as suggested.

Reviewer #2:

Thanks to the authors for answering my comments and making revisions where appropriate. I believe my comments have been fully addressed, and am happy to recommend the article for publication. Congratulations on an important piece of research

Thank you.

REVIEWERS' COMMENTS

Reviewer #1 (Remarks to the Author):

Thank you for addressing the comments. I have no further suggestions - and would be pleased to see this paper published.

Response to the referees' comments

Ms. Ref. No.: NCOMMS-21-41044C

Title: The atlas of unburnable oil for supply-side climate policies

We would like to thank the referees for the encouraging assessment of our manuscript and for the comments that they have provided.

REVIEWERS' COMMENTS

Reviewer #1:

Thank you for addressing the comments. I have no further suggestions - and would be pleased to see this paper published.

Thank you.